# A New High-Resolution Multi-Drought Indices Dataset for Mainland China

Qi Zhang, Chiyuan Miao*, Jiajia Su, Jiaojiao Gou, Jinlong Hu, Xi Zhao, Ye Xu

State Key Laboratory of Earth Surface Processes and Resource Ecology, Faculty of Geographical Science, Beijing Normal University, Beijing 100875, China

*Correspondence to*: C.Y.Miao (miaocy@bnu.edu.cn)

**Abstract.**   Drought indices are crucial for assessing and managing water scarcity and agricultural risks; however, the lack of a unified data foundation in existing datasets leads to inconsistencies that challenge the comparability of drought indices. This

study is dedicated to creating CHM_Drought, an innovative and comprehensive long-term meteorological drought dataset with a spatial resolution of 0.1° and data collected from 1961 to 2022 in mainland China. It features six pivotal meteorological drought indices: the standardized precipitation index (SPI), standardized precipitation evapotranspiration index (SPEI), evaporative demand drought index (EDDI), Palmer drought severity index (PDSI), self-calibrating Palmer drought severity index (SC-PDSI), and vapor pressure deficit (VPD), of which SPI, SPEI, and EDDI contain multi-scale features for periods of

2 weeks and 1–12 months. The dataset features a comprehensive application of high-density meteorological station data and a complete framework starting from basic meteorological elements (the China Hydro-Meteorology dataset, CHM). Demonstrating its robustness, the dataset excels in accurately capturing drought events across mainland China, as evidenced by its detailed depiction of the 2022 summer drought in the Yangtze River basin. In addition, to evaluate CHM_Drought, we performed consistency tests with the drought indices calculated based on Climatic Research Unit (CRU) and CN05.1 data and

found that all indices had high consistency overall and that the 2-week scale SPI, SPEI, and EDDI had potential early warning roles in drought monitoring. Overall, our dataset bridges the gap in high-precision multi-index drought data in China, and the complete CHM-based framework ensures the consistency and reliability of the dataset, which contributes to enhancing the understanding of drought patterns and trends in China. Free access to the dataset can be found at https://doi.org/10.6084/m9.figshare.25656951.v2 (Zhang and Miao, 2024).

**Keywords:** Drought monitoring; CHM_Drought; Drought dataset; China

## 1 Introduction

Drought is defined as a persistent shortage of water below normal levels, exerting various impacts on the functionality of natural ecosystems and socio-economic structures (WMO, 1986; WMO, 2009; WMO, 2002). It can significantly affect ecosystems (Gampe et al., 2021), agricultural practices (Lesk et al., 2021), water resources (Dobson et al., 2020), and socio-

economic conditions (Naumann et al., 2021). Between 1999 and 2020, droughts affected an average of 69.21 million people

annually worldwide, causing direct economic losses amounting to approximately \$62.7 billion (GNDAR, 2021). The progression of climate change foretells an increase in drought occurrences, escalating in frequency, intensity, duration, and scope (Wang et al., 2022a). Therefore, the development of high-quality, multi-index drought datasets has become crucial to monitor and analyze drought and reduce the losses caused by drought.

The diversity of drought types poses significant challenges in drought assessment, leading to their classification into four categories: meteorological, hydrological, agricultural, and socio-economic droughts (Mishra and Singh, 2010). Meteorological drought—originating primarily from insufficient precipitation and exacerbated by global warming effects like increased potential evapotranspiration (Aadhar and Mishra, 2020) and rising saturated vapor pressure differences (Gamelin et al., 2022)—is the foundational cause of other drought types (Zhang et al., 2022a). Considering the impact of agricultural drought

on both food crops and other vegetation types, some researchers have broadened its scope to encompass all natural and artificial vegetation, or even the entire ecosystem, termed ecological drought (Sadiqi et al., 2022). Hydrological drought is characterized by inadequate surface and groundwater resources within water resource management systems, with runoff data commonly utilized for its analysis (Dracup et al., 1980). Socio-economic drought, on the other hand, is associated with water resource systems' inability to satisfy water demands (Huang et al., 2016; Shi et al., 2018). Because meteorological drought is the initial

index and root cause of a series of interrelated drought types—such as agricultural drought, hydrological drought, and socio-economic drought—meteorological drought is the basis of drought research, and the meteorological drought index has the most subtypes in drought monitoring and quantification (Svoboda and Fuchs, 2017; Heim, 2002).

    The various meteorological drought indices each possess distinct advantages and limitations. The widely used Palmer drought severity index (PDSI) was an early metric; however, its applicability is limited in extreme climates and non-plains regions,

and this limitation led Wells et al. (2004) to develop the self-calibrating PDSI (SC-PDSI), which enhances spatial comparability by using dynamically calculated constants and region-specific calibration. Given the complexity of the PDSI calculation (the input data include precipitation, temperature, and available water content), the standardized precipitation index (SPI; McKee et al. 1993), which requires only precipitation data and is simple to calculate, is by far the most widely used index and features multiple timescales to account for the cumulative effects of drought. However, considering meteorological

drought's sensitivity to solar radiation, wind speed, air temperature, and relative humidity, Vicente-Serrano et al. (2010a) introduced the standardized precipitation evapotranspiration index (SPEI), which assesses droughts by calculating the climate water balance using the Penman–Monteith FAO equation (Allen et al., 1998) for potential evapotranspiration (PET). Building on the understanding of atmospheric factors influencing drought, the vapor pressure deficit (VPD) emerges as another crucial measure. The VPD quantifies the discrepancy between actual and saturated air moisture levels, with higher values signifying

more arid conditions (Gamelin et al., 2022). This metric adds value to drought analysis by representing the thirst of the atmosphere for moisture, a vital factor that many other drought indices do not consider. In addition, Hobbins et al. (2016) noted that most drought indices primarily focus on precipitation and temperature, with few directly reflecting evaporation dynamics. To address this, the evaporative demand drought index (EDDI) was established, using the relationship between atmospheric evaporation requirement ($E_0$; Allen et al., 1998) and actual evapotranspiration (AET), monitoring drought through

$E_0$'s response to surface drying anomalies. This exploration of drought indices highlights the need for high-quality drought data that reflect the various climatic factors that contribute to drought, and that such drought data are essential for accurately assessing drought and developing strategies that can mitigate its far-reaching effects.

     Global-scale drought datasets have been developed to assess and quantify the impacts caused by drought. The common ones mainly include global multi-scale SPEI calculated based on Climatic Research Unit (CRU) monthly meteorological data

(Beguería et al., 2010; Vicente et al., 2010), which spans the period 1901–2022 with a spatial resolution of 0.5° and covers the global land. Pyarali et al. (2022) also calculated SPEI, combining precipitation from the Climate Hazards group InfraRed Precipitation with Stations (CHIRPS) dataset and PET from the Global Land Evaporation Amsterdam Model (GLEAM), covering the period 1981–2018 at a spatial resolution of 5 km. Liu et al. (2024) combined European Centre for Medium-Range Weather Forecasts (ECMWF) Reanalysis v5 (ERA5) precipitation and PET developed by Singer et al. (2021) to produce a

multi-scale (5, 30, 90, 180, and 360 days) global SPEI dataset with a time span of 1982–2021 and a spatial resolution of 0.25°. In addition, there are some drought datasets such as SPEI calculated based on ERA5 data (Vicente-Serrano et al., 2023), PDSI calculated based on TerraClimate (Venkatappa and Sasaki, 2021), and PDSI and SPEI calculated on the basis of data from the Gravity Recovery and Climate Experiment (GRACE; Zhao et al., 2017). Since the accuracy of these datasets depends largely on the quality of the meteorological information, differences in the datasets used by different researchers to compute the same

indices can lead to considerable differences in the results, which complicates cross-sectional comparisons. This point highlights the urgent need to utilize consistent and high-quality meteorological datasets for the calculation of these indices. Also, most existing drought indices focus on monthly or longer timescales and may not capture short-term (e.g., weekly scale) meteorological drought conditions. In addition, there are still controversial aspects in the calculation methods of some indices, such as PET and reference crop evapotranspiration ($ET_0$), which have often been calculated in different ways. When

considering AET under energy or water constraints, the $ET_0$ estimates the upper limit of evapotranspiration under energy constraints, while under water constraints, the land–atmosphere feedback affects $ET_0$ in an opposite or complementary manner. Hobbins et al. (2016) suggested that $ET_0$ could serve as an independent drought indicator and developed EDDI. In contrast, Noguera et al. (2022) used PET, which is commonly used to calculate SPEI, instead of $ET_0$ to calculate EDDI; this approach may differ greatly from Hobbins et al. (2016) either conceptually or in terms of calculation results. Overall, addressing these

challenges requires a multifaceted approach that includes improving data quality and consistency, developing methods to capture a broader range of timescales, and clarifying drought index concepts and methods.

     Drought is one of the most important types of natural disasters in China, causing the loss of 10 million tons of grain production each year, and direct economic losses of up to 44 billion yuan per year (Su et al., 2018). Under global warming, the development of drought in China is showing a trend of increasing area, accelerating frequency, and worsening disaster (Zhang

et al., 2022b). Entering the 21st century, drought events became more frequent. In north China, northeast China, northwest China, and other areas, the drought situation is still severe, and some areas in the south have also become significantly drier with the increased frequency of major drought events (Zhai et al., 2010), especially the widespread drought in the summer of 2022 in China's Yangtze River basin. The evapotranspiration anomaly for the whole river basin in summer was the second

highest since 1960 (second only to 2013, with its high temperature and drought), which further aggravated the water deficit in the Yangtze River basin (Lyu et al., 2023).

Some scholars have developed drought datasets for China in order to better quantify, monitor, or forecast drought. Wang et al. (2021) developed daily versions of SPI and SPEI to quantify short-term meteorological droughts using data from 484 meteorological stations collected in the period 1961–2018, but their spatial coverage was limited to those 484 stations. There are also drought datasets calculated on the basis of different data products, such as Zhang et al. (2019), who integrated CN05.1 and near-real-time satellite precipitation products with the SPI dataset at a spatial resolution of 0.25° covering 1961–2016. Zhang et al. (2023a) constructed a daily-scale SPEI and SPI using data with a spatial resolution of 0.1° for the years 1979–2018 based on the China Meteorological Forcing Dataset (CMFD). Despite great progress in meteorological data sharing in China, high-resolution, multi-scale, multi-drought-index datasets are still lacking.

This paper aims to construct a new long-term (1961–2022) drought dataset (CHM_Drought), including SPI, SPEI, PDSI, SC-PDSI, and VPD. According to the characteristics of these indices, we also considered multiple timescales (among them, SPI, SPEI, and EDDI have 1- to 12-month and 2-week scales). Then we evaluated the performance of CHM_Drought after comparative validation and proved that CHM_Drought can accurately identify specific characteristics of drought in China and that the complete framework based on CHM can increase our understanding of the pattern and trend of drought in mainland China. This can provide strong support for the development of drought management and response strategies.

## 2 Datasets and processing

### 2.1 Data

We used several datasets, including the daily meteorological station data (Figure 1) from the China Meteorological Administration (CMA; http://data.cma.cn/), gridded precipitation data from CHM_PRE (Han et al., 2023; https://data.tpdc.ac.cn/zh-hans/data/e5c335d9-cbb9-48a6-ba35-d67dd614bb8c), and data from both CRU (https://crudata.uea.ac.uk/cru/data/hrg/) and CN05.1 (a gridded daily observation dataset over mainland China; https://ccrc.iap.ac.cn/resource/detail?id=228). First, we applied meteorological station data from the CMA to interpolate basic meteorological variables from 1961 to 2022 with spatial resolution of 0.1°, including maximum temperature (Tmax), minimum temperature (Tmin), mean temperature (Tmean), average wind speed (Wind), sunshine duration (Ssd), and average relative humidity (Rh). We directly used CHM_PRE and the interpolated meteorological data to compute CHM_Drought. The CN05.1 and CRU datasets were collected to evaluate CHM_Drought, with CRU data covering Precipitation (Pre), Tmax, Tmin, Tmean, Wind, and Ssd, and CN05.1 data covering Pre, Tmax, Tmin, Tmean, Wind, Rh, and Ssd. Notably, CN05.1's Ssd data spans 1961 to 2018, while other variables span 1961 to 2022.

In calculations for the drought index, as recommended by Li et al. (2023), we adopted the Global Gridded Surfaces of Selected Soil Characteristics data (https://daac.ornl.gov/cgi-bin/dsviewer.pl?ds_id=1006) for the soil available water capacity (AWC) data. When comparing with VPD data, we also used the third-generation normalized difference vegetation index (NDVI) of

the Global Inventory Monitoring and Modeling System (GIMMS; https://climatedataguide.ucar.edu/climate-data/ndvi-normalized-difference-vegetation-index-3rd-generation-nasagfsc-gimms) data for comparison, from 1982 to 2022. In evaluating the performance of CHM_Drought in the drought zone of China, we used the aridity index (AI)—that is, the ratio of annual precipitation to potential evapotranspiration—to classify the arid regions of China (Figure 1) (Li et al., 2021; https://csidotinfo.wordpress.com/data/global-aridity-and-pet-database/).

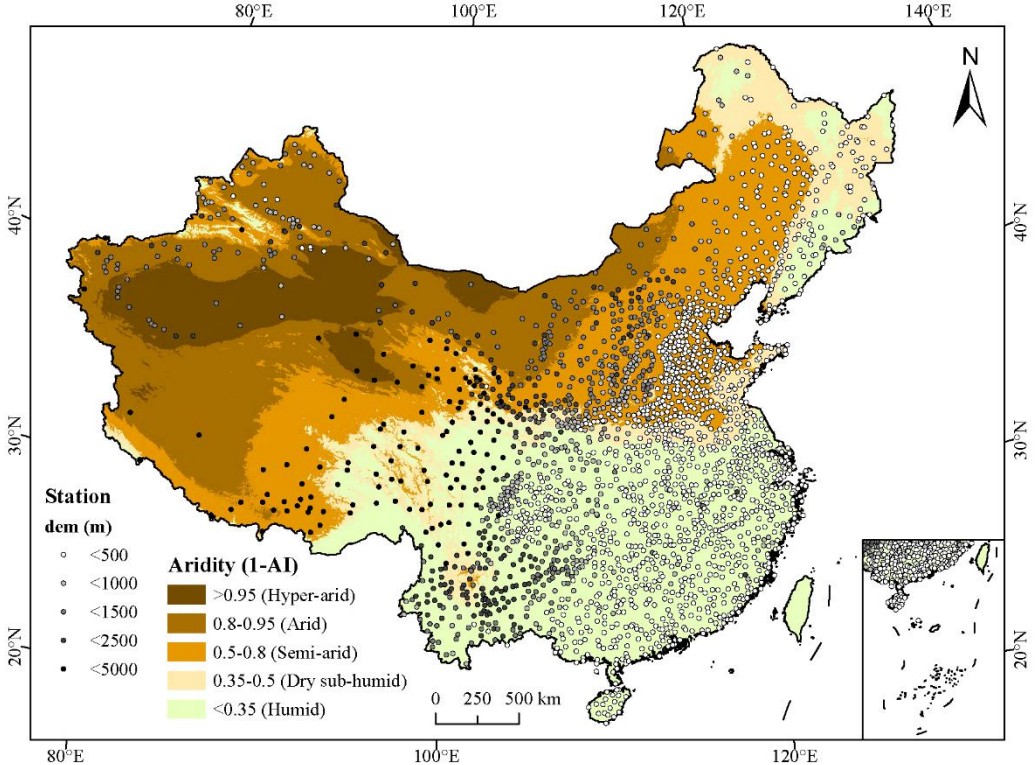

**Figure 1: Distribution of 2,419 meteorological stations and pattern of China's drylands.** The drylands are further classified into four subtypes based on the aridity $(1 - AI)$: hyper-arid $(AI < 0.05)$, arid $(0.05 \leq AI < 0.20)$, semi-arid $(0.20 \leq AI < 0.50)$, and dry sub-humid $(0.50 \leq AI < 0.65)$. Shaded circles mark stations, and darker circles indicate higher elevations (digital elevation model, DEM).

## 2.2 Data processing

To ensure the integrity and reliability of our dataset, rigorous data quality control measures were implemented during the preprocessing stage. This involved a comprehensive data-cleaning process to address various aspects, including the identification and treatment of outliers and handling of missing values. First, outliers within the meteorological station data were identified and addressed using appropriate statistical techniques. This step aimed to detect any data points that deviated significantly from the expected distribution and could potentially distort the analysis results. Second, missing values present in the dataset were carefully handled to minimize their impact on the overall dataset quality: we have removed any missing values to ensure that only valid sites are used for daily data interpolation.

Before calculating the drought index, we interpolated the basic meteorological variables (Tmax, Tmin, Tmean, Wind, Ssd, Rh; see Figure 2), incorporating the correlation decay distance (CDD) specific to each variable (Figure S1). Details on CDD are provided in the supplementary document. For the  interpolation process, we adopted angular distance–weighted interpolation (ADW), which considers angular weight in addition to the distance weight function, making it more robust to outliers. ADW with CDD provides a key benefit that other methods may not emphasize as directly: the gradual decrease of correlation with increasing distance between stations. For missing values, we did not fill in the time series, but used only stations with available data for spatial interpolation each day. We interpolated meteorological elements to 0.1° spatial resolution, which is consistent with CHM_PRE.

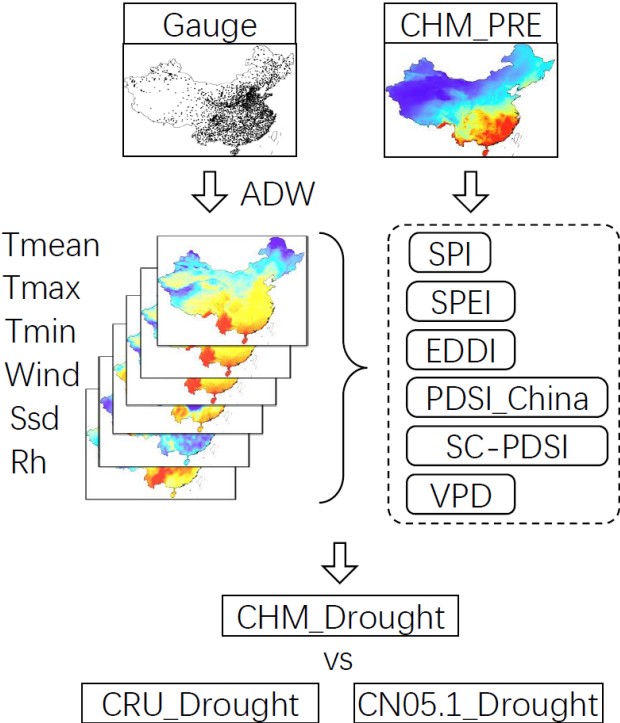

**Figure 2: Flowchart of the drought index construction system.** The meteorological variables include maximum temperature (Tmax), minimum temperature (Tmin), average temperature (Tmean), average wind speed (Wind), sunshine duration (Ssd), and average relative humidity (Rh). CRU_Drought and CN05.1_Drought represent drought indices calculated based on CRU and CN05.1 meteorological data, respectively.

**Table 1. CHM_Drought dataset summary table, including drought index calculation input variables, timescale, and index characteristics.**

| Drought index | Input parameters | Timescale | Characteristics |
|---|---|---|---|
| SPI | P | 2 weeks, 1–12 months | The calculation is simple and widely used |

| | | | |
|---|---|---|---|
| SPEI | P, PET | 2 weeks, 1–12 months | Similar to SPI but with a temperature component |
| EDDI | ET0 | 2 weeks, 1–12 months | EDDI shows the anomaly in evaporative demand aggregated |
| PDSI_China | P, PET, AWC | month | Parameter calibration with Chinese regional characteristics |
| SC-PDSI | P, PET, AWC | month | SC-PDSI is developed for each station/grid and changes based upon the climate regime of the location |
| VPD | Tmean, Rh | month | VPD affects the closure of plant stomata and describes how dry the air is |

Note: All abbreviations: P, precipitation; ET0, reference crop evapotranspiration; PET, potential evapotranspiration; Tmean, average temperature; Rh, average relative humidity; AWC, soil available water capacity.

## 3 Methodology

### 3.1 Standardized precipitation index (SPI)

SPI is a dryness index proposed by American scholars McKee et al. (1993) and used to analyze the drought situation in Colorado. It is a powerful, flexible, and simple index, which takes precipitation as the research object, monitors precipitation on a long timescale, characterizes the correlation between precipitation and climate characteristics within a certain period of time, and is as effective for the analysis of wet periods or cycles as for the analysis of dry periods or cycles. Because SPI has the characteristics of multiple timescales, these timescales can reflect the impact of drought on the availability of different water resources. SPI is to calculate the distribution probability, $\Gamma$, of precipitation within a certain period of time, then to perform normal standardization, and finally to classify the drought level with the standardized precipitation cumulative frequency distribution:

$$f(x) = \frac{1}{\beta^\gamma \Gamma(\gamma)} x^{\gamma-1} e^{-x/\beta} \quad x > 0 , \tag{1}$$

where $\beta > 0$ and $\gamma > 0$ are scale and shape parameters, respectively. The detailed calculation method can be found in the supplementary material.

### 3.2 Standardized precipitation evapotranspiration index (SPEI)

Compared with SPI, SPEI more comprehensively reflects the relationship between precipitation and potential evapotranspiration (PET) and better reveals the impact of the hydrological cycle (Vicente-Serrano et al., 2010a). Since SPEI considers the sensitivity of atmospheric evaporation demand to drought, it is especially suitable for dry and warm climate

zones in areas with increased temperature and PET and can better capture drought dynamics than SPI (Li et al., 2020). First, the PET is calculated. The second step is to calculate the difference between precipitation (P) and PET, $D = P - PET$. The third step is to transform data $D$ as SPI:

$$SPEI = W - \frac{(c_2 W + c_1)W + c_0}{[(d_3 W + d_2)W + d_1]W + 1},$$

(2)

where $W = \sqrt{-2\ln(T)}$, T is the probability of a definite D value, values of coefficients are follows: $c_0 = 2.515517$, $c_1 = 0.802853$, $c_2 = 0.010328$, $d_1 = 1.432788$, $d_2 = 0.189269$, and $d_3 = 0.001308$. The detailed calculation method can be found in the supplementary material. According to the FAO standard, we use the Penman equation to calculate PET as follows:

$$PET = \frac{0.408\Delta(R_n - G) + \gamma \frac{900}{T+273} u_2(e_s - e_a)}{\Delta + \gamma(1 + 0.34u_2)},$$

(3)

where $\Delta$ is the slope of the saturated vapor pressure–temperature relationship (kPa $\cdot$ °C$^{-1}$), $R_n$ is the net radiation at the ground surface (MJ $\cdot m^{-2}d^{-1}$), and $G$ is the soil heat flux (MJ $\cdot m^{-2}$); on a timescale of 1 to 10 days, the soil heat capacity of the reference meadow is quite small and can be neglected. $\gamma$ is the psychrometric constant (kPa $\cdot$ °C$^{-1}$), T is the mean daily air temperature at 2-m height (°C), $u_2$ is the wind speed at 2-m height (m $\cdot s^{-1}$), $e_s$ is the saturation vapor pressure of the air (kPa), and $e_a$ is the actual vapor pressure of the air (kPa). The detailed calculation formulas for $\Delta$, $\gamma$, $e_a$, $e_s$, and $R_n$ can be found in the supplementary material.

### 3.3 Evaporative demand drought index (EDDI)

EDDI was developed by Hobbins et al. (2016) as an indicator of atmospheric drying potential, which can indicate plant stress on the ground. Therefore, the physically based ET$_0$ index has the advantage of being more direct and more dependent on atmospheric physics principles than are the SPEI and AET calculation methods that rely on remote sensing data. EDDI, similar to SPI and SPEI, incorporates multiple timescales, and the accumulation time can vary from 1 week to 1 year or more. For the calculation of ET$_0$ in EDDI, we used the standardized reference evapotranspiration equation (Allen et al. 2005) adopted by the American Society of Civil Engineers (ASCE) in developing the EDDI. Although some scholars have equated PET and ET$_0$ in recent years (Noguera et al., 2022), there are differences between the two (Xiang et al., 2020).

$$ET_0 = \frac{0.408\Delta(R_n - G) + \gamma \frac{C_n}{T+273} u_2(e_s - e_a)}{\Delta + \gamma(1 + C_d u_2)},$$

(4)

where $C_n$ (K mm s$^3$ Mg$^{-1}$ day$^{-1}$) and $C_d$ (s m$^{-1}$) are the "numerator constant" and "denominator constant," respectively, with values defined in Allen et al. (2005). EDDI is derived using the inverse method approximation detailed in Vicente-Serrano et al. (2010), which is repeated here for convenience:

$$EDDI = W - \frac{(c_2 W + c_1)W + c_0}{[(d_3 W + d_2)W + d_1]W + 1},$$

(5)

Values of coefficients are as follows: $c_0 = 2.515517$, $c_1 = 0.802853$, $c_2 = 0.010328$, $d_1 = 1.432788$, $d_2 = 0.189269$, and

$d_3 = 0.001308$.

For $P(ET_0) \leq 0.5$,

$$W = \sqrt{-2\ln(P(ET_0))} ,\qquad(6)$$

and for $P(ET_0) > 0.5$,

$$W = \sqrt{-2\ln(1 - P(ET_0))} .\qquad(7)$$

Please refer to the supplementary material for a more detailed description of EDDI.

## 3.4 Palmer drought severity index (PDSI_China)

The PDSI is a drought index with clear physical meaning established by Palmer (1965). Its introduction was an important turning point in the history of drought index research. PDSI is one of the most widely used drought indices in meteorological drought research and monitoring (Aiguo et al., 2004). When calculating water balance, PDSI considers pre-season precipitation

and water supply and demand, with clear physical meaning. Water deficit ($d$) is the difference between actual precipitation ($P$) and climate-appropriate precipitation ($P'$). To make the PDSI a standardized index, after the water deficit $d$ is determined, it is multiplied by the climate weight coefficient $K$ of a given month in a given place to obtain the water anomaly index $Z$, also known as the Palmer $Z$ index, which indicates the deviation degree between the actual climate dry–wet condition and its average water condition in a given month and place: $Z = dK$; the value of $K$ is determined by the month and geographical

location:

$$K_i = \frac{a}{\sum_{j=1}^{12} \overline{D_J} K_j'} K_i' .\qquad(8)$$

The empirical constant $a = 17.67$ obtained by Palmer from the data of nine stations in seven states was revised to 16.84 according to the climate characteristics of China (Zhong, 2019); therefore, we calculate it as PDSI_China, where $\sum_{j=1}^{12} \overline{D}_J K_j'$ is the average annual absolute moisture anomaly over the years, and $j$ represents January to December. The methodology is

described in the supplementary document.

## 3.5 Self-calibrating Palmer drought severity index (SC-PDSI)

Based on PDSI, Wells et al. (2004) proposed and evaluated an SC-PDSI. SC-PDSI automatically calibrates the index behavior at any location by replacing the empirical constants in the index calculation with dynamically calculated values. Compared with PDSI, it can adapt to local climate (Dai, 2011) and has been proven to have better applicability in China (Bai et al., 2020;

Shao et al., 2018). Since the disadvantages of PDSI mainly revolve around its inconsistency between different locations, and it uses multiple empirical parameters that depend on the study area in the calculation process, Wells et al. (2004) believed that

changing the ratio ($\widetilde{K}$) could solve the spatial inconsistency of PDSI without changing the way PDSI deals with climate seasonal changes.

$$\widetilde{K} = \frac{a}{\sum_{j=1}^{12} \overline{d_j} K'_j} K'_i \ . \tag{9}$$

Since $\sum_{j=1}^{12} \overline{d_j} K'_j$ can be approximately regarded as the annual sum of the average absolute value of $Z$ ($\widetilde{Z} = \sum_{j=1}^{12} \overline{d_j} K'_j$), and the value of $a$, 17.67 as obtained by Palmer, is the average value of $\widetilde{Z}$ (i.e., the annual average sum of vapor anomalies), and since PDSI is based on cumulative vapor anomalies, so $\widetilde{K} = \frac{expected\ average\ PDSI}{observed\ average\ PDSI}$. The non-extreme value range of PDSI is defined as −4 to 4, but in practice this range is different. Palmer (1965) argues that if the PDSI were truly a standardized measure of drought severity, then values outside of that range (−4 to 4) would occur with roughly the same frequency. If the

frequency of extreme events is $f_e$, then the $f_e$th percentile should be −4.00 and the $(100 - f_e)$th percentile should be 4.00. So $\widetilde{K} = \frac{expected\ feth\ percentile\ of\ the\ PDSI}{observed\ fe\ percentile\ of\ the\ PDSI}$. Defining an extreme drought as a "one in 50 year event" does not determine the percentage of PDSI values below −4.00, as it may last two months or two years. In this implementation, Wells et al. (2004) used an $f_e$ value of 2%, which resulted in the following climate characterization equation:

$$K = \begin{cases} K'(-4\ /\ 2nd\ percentile), if\ d < 0 \\ K'(4\ /\ 98th\ percentile), if\ d \geq 0 \end{cases} . \tag{10}$$

See the supplementary materials for detailed formulas.

### 3.6 Vapor pressure deficit (VPD)

Vapor pressure deficit (VPD) is one of the most important climate variables used to simulate the flux and state of water and carbon in ecosystem models, and one of the main driving factors of vegetation evapotranspiration. It is also an important meteorological variable in fire warning models and warning models for the spread of pests, diseases, and epidemic diseases

(Green and Hay, 2002). Therefore, VPD is widely used in various hydrological cycle, vegetation carbon cycle, and evapotranspiration estimation models (Hashimoto et al., 2008; Wang and Dickinson, 2012). Saturated vapor pressure is a function of temperature and can be directly calculated from temperature, as shown in the Tetens empirical formula (Allen et al., 1998):

$$e^0(T) = 0.6108 exp\left[\frac{17.27T}{T+237.3}\right], \tag{11}$$

where $T$ is the air temperature (°C) and $e^0(T)$ is the saturated water vapor pressure at temperature (kPa). Since the above equation is a nonlinear function, for the average saturated vapor pressure with such a long interval at the monthly scale, if the average temperature is used to replace the daily maximum and minimum temperatures, the estimated value of the average saturated vapor pressure will be low, and the corresponding vapor pressure difference will be small. Therefore, the mean value

of the saturated vapor pressure corresponding to the daily average maximum and minimum temperatures within the time interval is used for calculation (Li et al., 2014):

$$e_s = \frac{e^0(T_{max}) + e^0(T_{min})}{2},$$
(12)

where $e_s$ is the average saturated vapor pressure (kPa), and $T_{max}$ and $T_{min}$ are the daily average highest and lowest air temperature (°C), respectively. The actual vapor pressure $e_a$ (kPa) is calculated according to the monthly average relative humidity ($\varphi_{mean}$): $e_a = e_s \frac{\varphi_{mean}}{100}$, and VPD $= e_s - e_a$.

### 3.7 Consistency test for the drought dataset

To evaluate the consistency of CHM_Drought, we calculated the same index, namely CRU_Drought and CN05.1_Drought, using CRU and CN05.1 data, respectively (see section 2). For consistency testing of the CHM_Drought (the data characteristics are shown in Table 1), we resampled both the CHM_Drought (0.1°) and CN05.1_Drought data (0.25°) to 0.5° to match the spatial resolution of CRU_Drought (0.5°).

Pearson's correlation coefficient (CC) and Nash-Sutcliffe efficiency coefficient (NSE) were used as the evaluation indices of data consistency to detect the consistency of CHM_Drought, CN05.1_Drought, and CRU_Drought with the same spatial resolution (0.5°) and the same time span (1961–2022). To assess the differences in the consistency of different timescales, we selected 1-, 3-, 6-, and 12-month scales for evaluation; the results are presented in section 4.2. The formulas are as follows:

$$NSE = 1 - \frac{\sum_{i=1}^{N}(y_i - \widehat{y_i})^2}{\sum_{i=1}^{N}(y_i - \bar{y})^2},$$
(13)

$$CC = \frac{\sum_{i=1}^{N}(x_i - \bar{x})(y_i - \bar{y})}{\sqrt{\sum_{i=1}^{N}(x_i - \bar{x})^2}\sqrt{\sum_{i=1}^{N}(y_i - \bar{y})^2}},$$
(14)

where $\bar{y} = \frac{\sum_{i=1}^{N} y_i}{N}$, $y_i$ is the CHM_Drought value at time $i$ ($i = 1, \cdots, N$), $\bar{y}$ is the mean value taken over N, N is the total data size of $y_i$ ($i = 1, \cdots, N$), and $\widehat{y_i}$ is the CRU_Drought (or CN05.1_Drought) value at time $i$; $\bar{x}$ and $\bar{y}$ represent the sample means of the two, respectively.

## 4 Results and discussion

### 4.1 Performance of CHM_Drought during the 2022 summer drought in the Yangtze River basin

A severe drought occurred in the south of China in the summer of 2022, mainly concentrated in the Yangtze River basin. To show the performance of the CHM_Drought dataset in monitoring drought, we use the summer (June, July, August; JJA) of 2022 in the Yangtze River Basin as an example to examine the monitoring capabilities of drought indices. For SPI, SPEI, and EDDI we selected a 3-month scale (seasonal scale; Jin et al., 2020), as shown in Figure 3. Due to the cumulative effect of

drought, the drought performance is different on different timescales. Therefore, we also examined the 2-week and 1-, 3-, and 6-month scales (Figure 4), in which the 2-week scale takes mid-August 2022 as the node, while the 1-, 3-, and 6-month scales all show the value from August 2022.

The indices exhibit varied degrees of drought severity, with each index offering a unique perspective based on its inherent parameters. For instance, SPI-3 and SPEI-3, focusing on precipitation anomalies, highlight significant deficiencies across the

central and eastern regions, aligning with the Yangtze River basin's experiences. EDDI-3, which emphasizes evaporative demand, suggests a widespread and intense drought condition, notably in the southern regions, indicating a profound hydrometeorological imbalance; this is similar to the 2022 summer high temperature profile shown by Ma et al. (2023). The PDSI_China and SC-PDSI indices, which incorporate soil moisture conditions and long-term climatic context, reveal extreme drought severity levels in the Yangtze River basin. These conditions reflect the compound effects of prolonged precipitation

deficits, high temperatures, and the resulting soil moisture depletion. Lastly, the VPD index maps out the atmospheric moisture demand, which reached anomalously high levels in the illustrated period, particularly in the Yangtze River basin. This condition aggravates the drought impact by enhancing evapotranspiration rates, which in turn further depletes soil moisture and stresses vegetation. Wang et al. (2023) have also demonstrated that a record-breaking compound drought–heatwave hit the Yangtze River basin in summer 2022, resulting in the strongest anomalies of VPD and soil moisture since 2000.

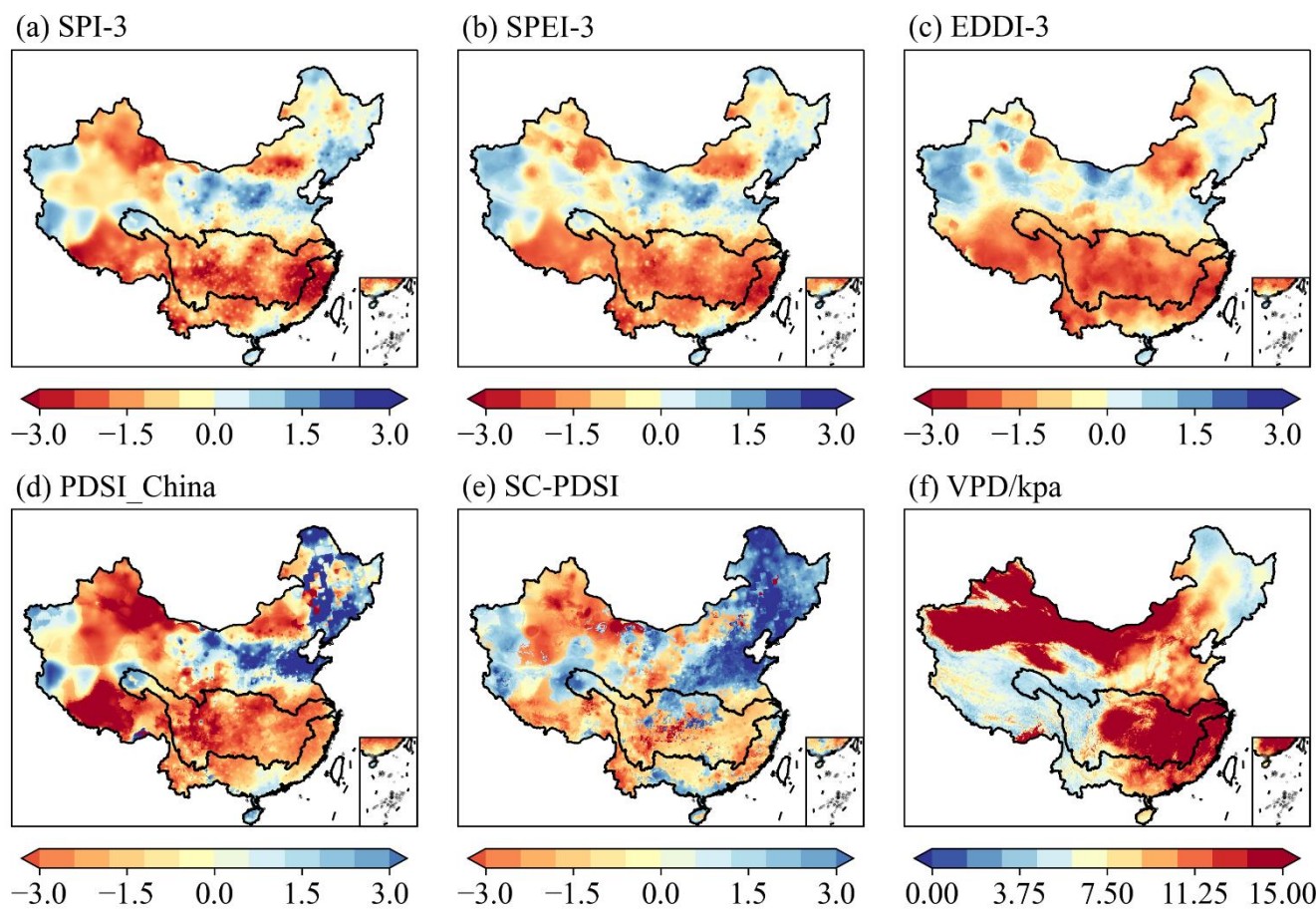

**Figure 3: Spatial distribution of summer (June, July, August; JJA) drought characteristics in the Yangtze River Basin, China.** Here, (a), (b), and (c) depict the three-month scale spatial distribution of drought indices, while (d), (e), and (f) present the average summer (JJA) values for these indices.

On different timescales of the same index, different information emerges. At the 2-week scale (Figure 4) the indices provide a snapshot of the immediate drought situation, which is highly valuable for short-term drought relief and response planning. Over 1-month and 3-month scales, the indices begin to show patterns of persistent drought conditions; these scales are critical for assessing the medium-term impact on agriculture and water resources. The 6-month scale reveals long-term drought conditions, which are crucial for planning and managing water resources, as well as in understanding the broader environmental impacts of extended droughts. It is also informative to compare different indices at the same timescale. We found that the results of SPI-2W (Figure 4a) show that the southern part of China, mainly the Yangtze River basin (Zhang et al., 2023b), has a short-term precipitation gap, and the precipitation in this region is far below the average level. SPEI-2W (Figure 4e) not only reflects the lack of precipitation but also takes into account the possible increase in evaporation due to high temperatures, making the drought in the southern region more severe. The effects of temperature increase that may not be captured by SPI are reflected in SPEI. For EDDI-2W (Figure 4i), the dramatic increase in atmospheric water demand is a direct result of the

heat wave, and EDDI-2W shows that the whole of China, and especially the Yangtze River basin, is in this drying condition, which may further exacerbate soil drying and water stress on crops.

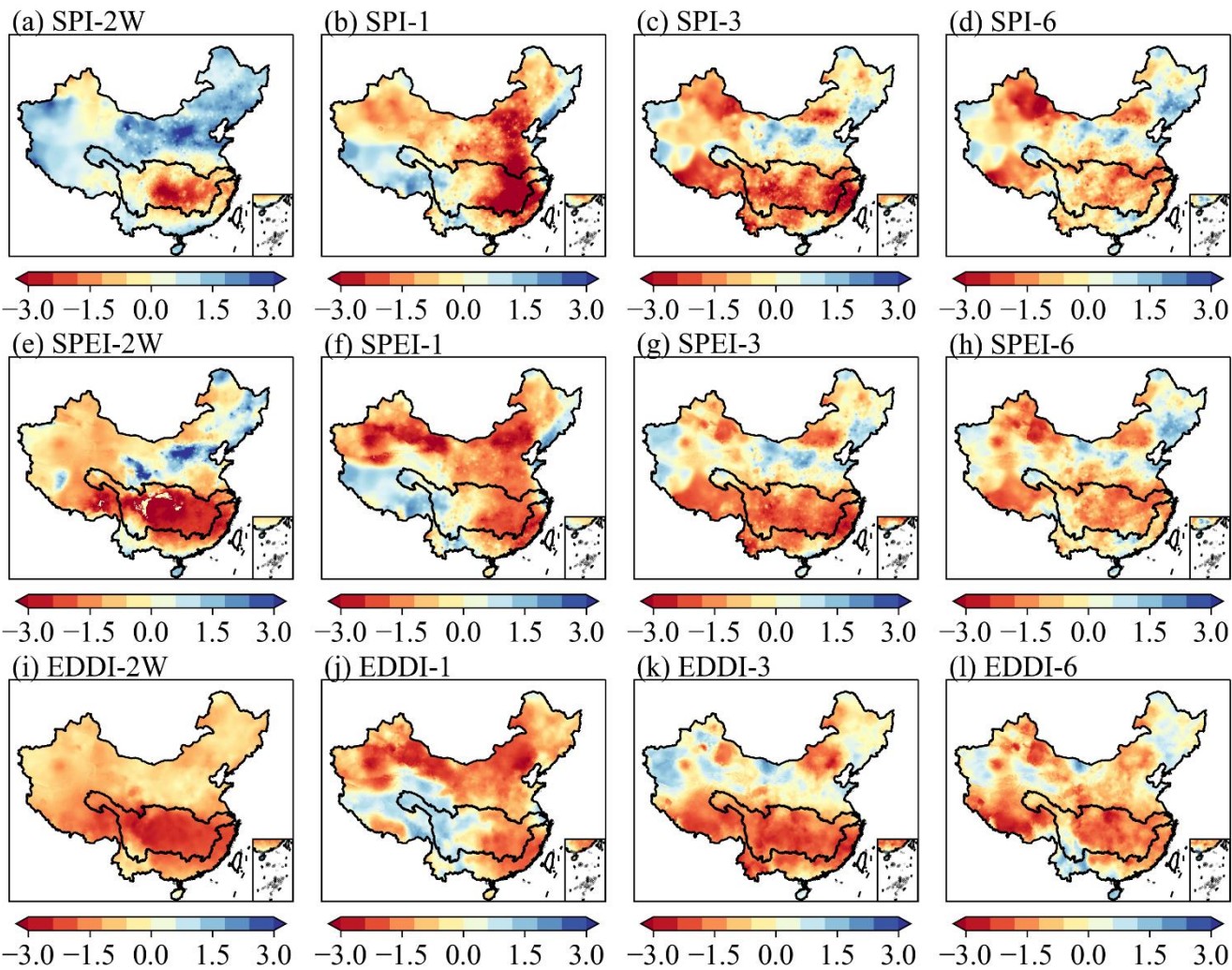

**Figure 4: Spatial distribution of three drought indices (SPI, SPEI, and EDDI) in the Yangtze River Basin, China, across multiple timescales (2-week, 1-month, 3-month, and 6-month), using August 2022 as an example.** (a–d) SPI-2W indicates the 2-week scale SPI,
SPI-1 indicates the 1-month scale SPI, SPI-3 indicates the 3-month scale SPI, and SPI-6 indicates the 6-month scale SPI. The scales of SPEI and EDDI follow the same naming pattern.

## 4.2 Consistency assessment of multi-scale SPI, SPEI, and EDDI based on CHM_Drought with CRU_Drought (or CN05.1_Drought)

Figure 5 illustrates the spatial distribution of CC values based on CHM for SPI-6, SPEI-6, and EDDI-6 with those calculated
based on CRU and CN05.1, respectively. Figure S2 is similar to Figure 5 but illustrates the spatial distribution of NSE. We can see that the correlations at the 6-month scale are high overall, with the correlations above 0.8 in most regions, especially

in the wet areas at low altitudes, and the correlations in the northwest region are generally lower than those in the southeast region, especially in the Qinghai–Tibet Plateau region, which has the lowest station density (Miao et al., 2024). However, most of the data developed so far are limited by the poor performance of sparse sites, whether they are developed meteorological

data (He et al., 2020; Wu and Gao, 2013) or drought datasets (Wang et al., 2021). Simply comparing the three indices, the overall correlations and NSE values of SPI and EDDI are generally higher than those of SPEI, while the regions with low correlations and NSE values of SPEI (Figure 5 and Figure S2) are mainly concentrated in the extremely arid regions. One consideration is that the correlations and NSE values depend on the quality and accuracy of the dataset. Another consideration is that, for the arid regions, the measurement of precipitation and evaporation is more difficult than it is in the wet regions, and

SPEI is not a single meteorological input compared with SPI and EDDI, but a series of "P − PET" values, resulting in greater noise (uncertainty) in the data.

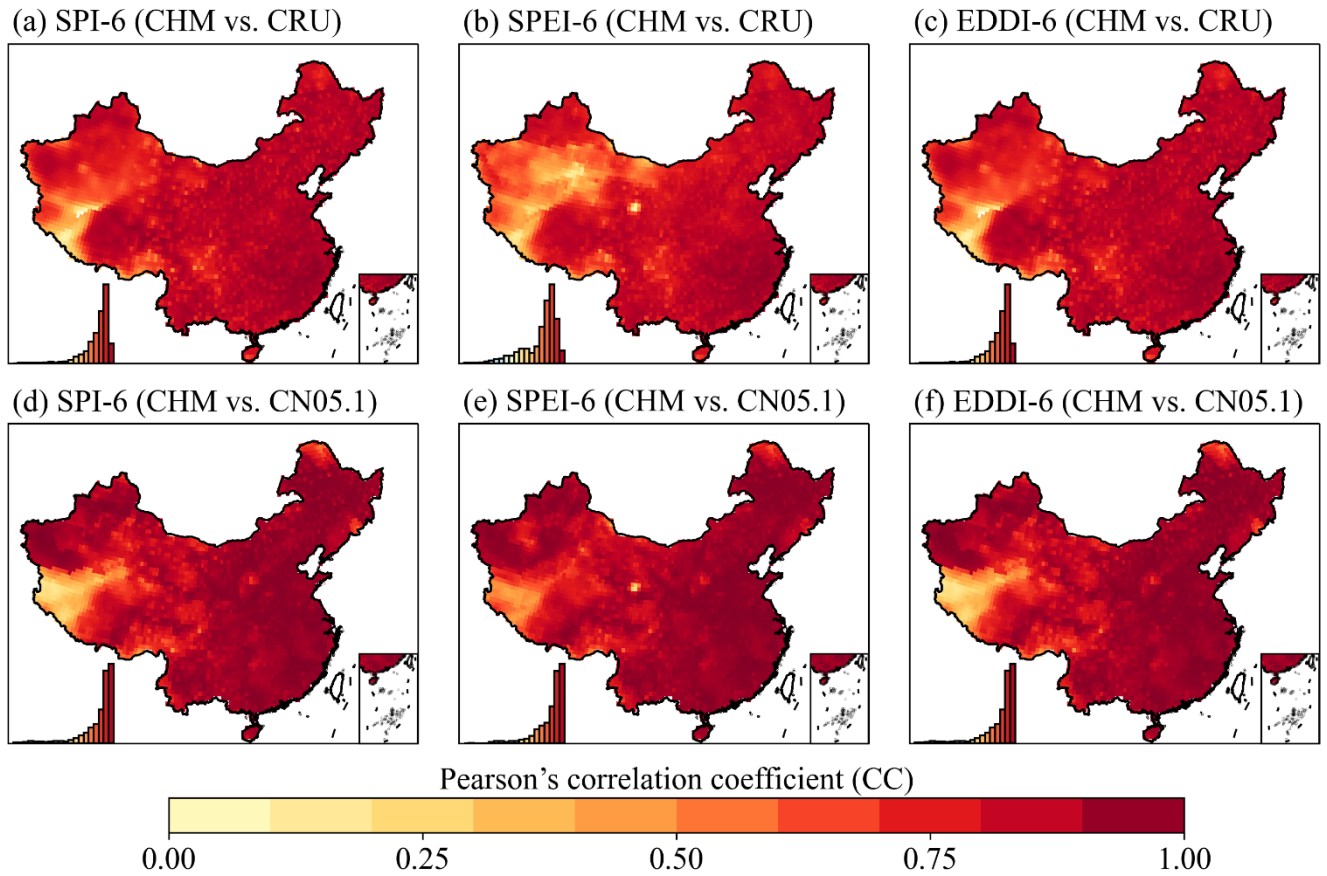

**Figure 5: (a–c) Correlation spatial distributions of SPI-6, SPEI-6, and EDDI-6 based on CHM and CRU data. (d–f) Correlation spatial distributions of SPI-6, SPEI-6, and EDDI-6 based on CHM and CN05.1 data.** The histogram at the bottom left in each subplot

shows the distribution of correlation coefficients for all grid cells.

When the timescale is changed, for example to a longer timescale (12 months; from Figures S2, S3, and S4), we find that the overall correlation between CHM_Drought and CRU_Drought (or CHM_Drought and CN05.1_Drought) remains high, which

proves the robustness of the data. But the spatial distributions of CHM_Drought vs. CRU_Drought and CHM_Drought vs. CN05.1_Drought still differ in the northwest; when compared with CN05.1_Drought, the regions with large differences are

on the west side of the Qinghai–Tibet Plateau, where the sites are notably the sparsest (Figure 1); when compared with CRU_Drought, the region with poor consistency also includes the hyper-arid region (Figure 1).

Consistency assessment at different timescales is shown in Figure 6. With an increase in timescale, although the median value in the box plot remains basically unchanged, the lower quartile shifts downward progressively. This indicates that the consistency of the SPEI calculated by the two datasets decreases as the timescale increase, especially in areas with low

correlation (such as the arid northwest region). As can be seen from Figure S2 and Figure S4, at the 12-month scale the NSE value in the northwest arid region is lower than it is at the 6-month scale. Aside from the limitations of the observational data, this may be due to climate variability, as climate factors (such as precipitation patterns and drought frequency and intensity) may have greater changes, resulting in larger inconsistency in the subsequent long-term records. It is also possible that in the arid region, extreme climate events (such as extreme drought or rainstorm) may occur more frequently, and these extreme

events may increase inconsistency at a long timescale. As can be seen from Figure 6c and 6d, the inconsistency between SPEI and SPI and EDDI is the largest at the monthly scale, but the consistency increases with the increase in timescale. This may be due to the accumulation of precipitation and evaporation processes over time: evaporation (or potential evapotranspiration) is a dynamic process that takes time to accumulate in sufficient quantities, as is precipitation. In the short term, some extreme weather events (such as heavy rain or drought) may affect the amount of precipitation or evaporation, but in the long term such

events may have only a small effect. Therefore, on a longer timescale, SPI and SPEI may reflect a more consistent and stable drought condition rather than being affected by short-term weather events.

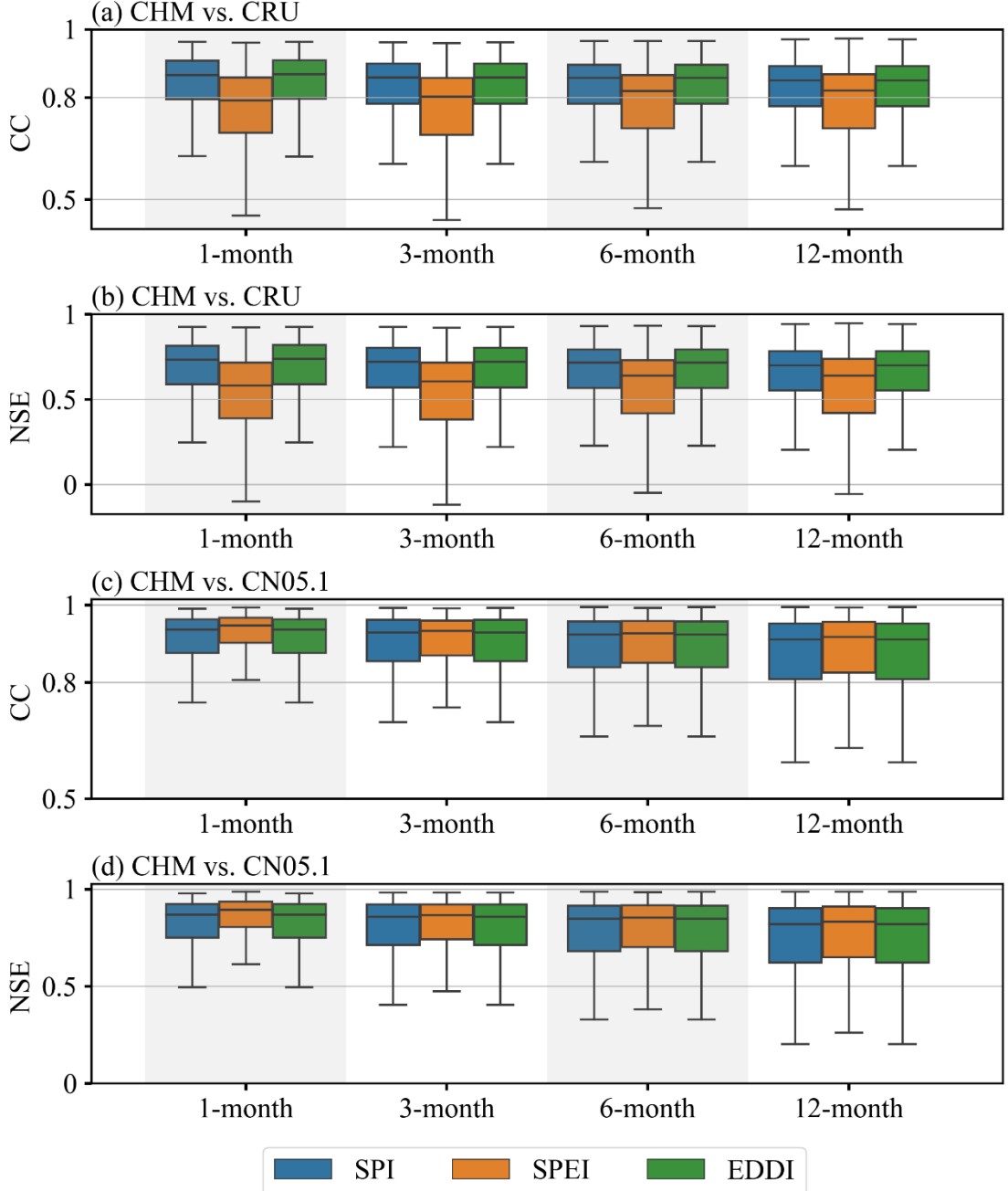

**Figure 6: Box plots of the CC and NSE of three drought indices (SPI, SPEI, EDDI) calculated based on CHM and either (a, b) CRU**
**or (c, d) CN05.1 at different month scales (1-, 3-, 6-, and 12-month).** The middle line within each box indicates the median, the upper and the lower edges mark the 25th and 75th percentiles, and the whiskers show the 2.5th and 97.5th percentiles.

In addition, we quantified the uncertainties of SPI, SPEI and EDDI at different time scales (Figure 7). We used standard deviation to quantify the results, which were similar to those in Figures S2–S4. The regions with higher standard deviation,

such as the arid northwest, highlight the spatial variability in uncertainty across different datasets. This suggests that the

drought indices calculated from these datasets may show obvious discrepancies in regions with sparse observational coverage. The observed uncertainties can be attributed to several factors: (1) interpolation techniques. The variability in interpolation techniques across datasets is a critical factor contributing to uncertainty. For instance, the CHM dataset employs advanced interpolation techniques based on high-density observational stations, while the CRU and CN05.1 datasets utilize thin plate smooth spline (TPSS) and inverse distance weighting (IDW) methods, respectively (Harris et al., 2020; Xu et al., 2009). These

methodological differences become particularly pronounced in areas with complex topography, such as the arid northwest. Xu et al. (2022) demonstrated that TPSS performs well in capturing broad climate gradients, it may overly smooth the results in data-sparse regions, leading to underestimation of extremes. Conversely, IDW might overemphasize local station values, causing biases in interpolated fields (Shen et al., 2023). (2) Sparse observational coverage. Limited observational inputs in certain regions further exacerbate uncertainty. Liu et al. (2009) highlighted that the density of interpolation sites is the key

factor influencing interpolation accuracy. They found that the performance of interpolation methods, such as kriging or IDW, deteriorates significantly as the number of sites decreases. (3) Inclusion of auxiliary covariates. Differences in the incorporation of auxiliary covariates, such as topography, land cover, or climate zones, also contribute to dataset discrepancies. For instance, the CHM dataset incorporates high-resolution digital elevation models (DEM) as covariates, while the CRU dataset primarily relies on planar spatial gradients, without explicitly considering terrain effects (Harris et al., 2020). This leads to substantial

differences in regions with complex orography.

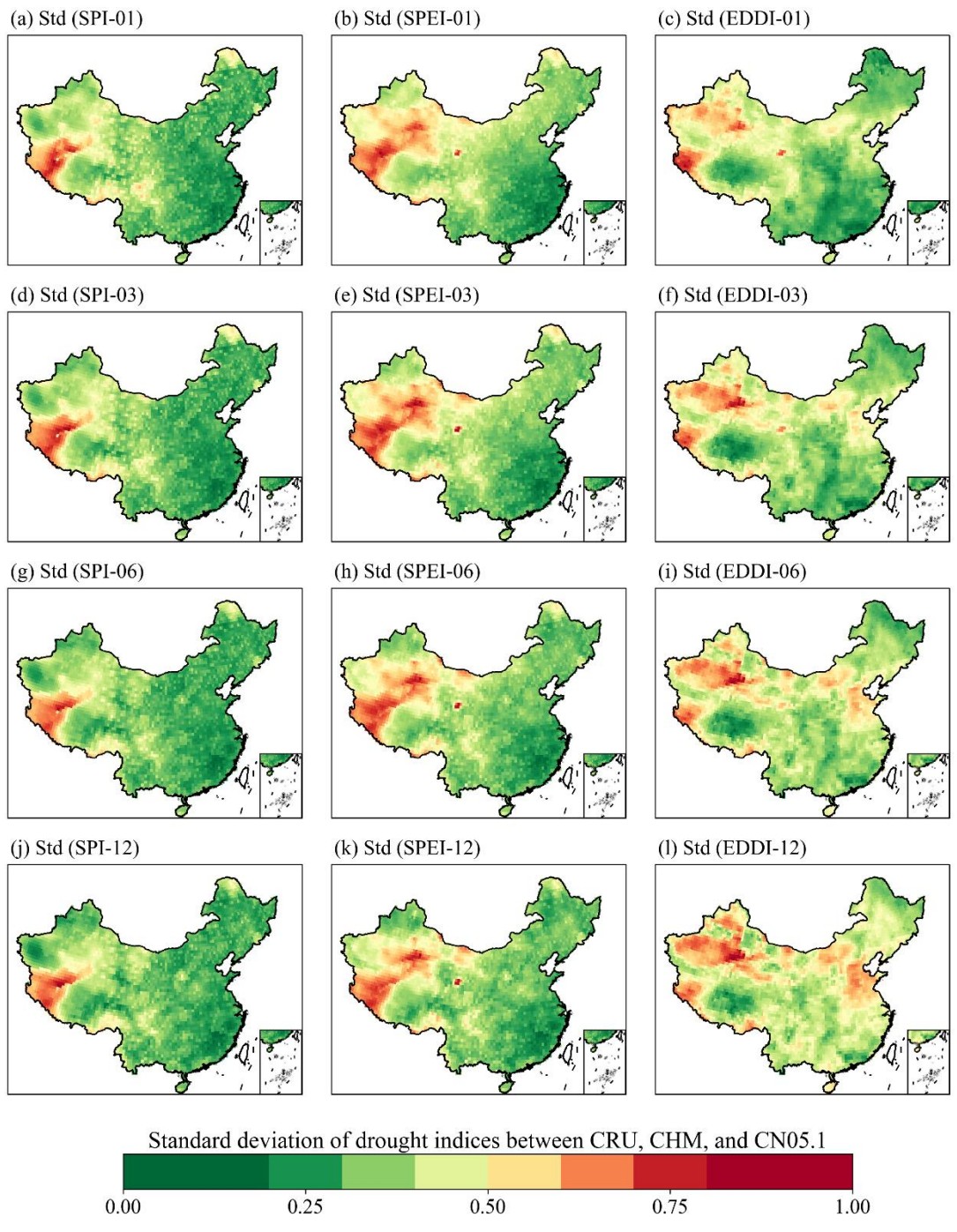

**Figure 7: Spatial distribution of the standard deviations of the SPI, SPEI, and EDDI drought indices across three data sources (CRU, CHM, and CN05.1) at various time scales (1-, 3-, 6-, and 12-month).** Here, (a–c) show the 1-month scale, (d–f) show the 3-month scale, (g–i) show the 6-month scale, and (j–l) show the 12-month scale.

### 4.3 Consistency assessment of PDSI_China and SC-PDSI based on CHM_Drought with CRU_Drought (or CN05.1_Drought)

We evaluated the consistency of PDSI_China and SC-PDSI in China. According to Figure 8 and Figure S5, the two indices have high correlations over China as a whole. However, PDSI_China corrected according to data from Chinese meteorological stations is significantly better than SC-PDSI. Except for the areas with low station density, the overall correlation is high, especially in the wet areas. The biggest difference between PDSI_China and SC-PDSI comes from the calibration method. SC-PDSI uses the self-calibration method, but it may not consider the regional differences in China, which may affect the accuracy of SC-PDSI, because it relies on appropriate calibration to reflect the climate characteristics of specific areas. In addition, SC-PDSI takes into account the climate characteristics of specific areas through the self-calibration method, making it, in theory, more sensitive to local climate change. Therefore, although it does not have multi-scale characteristics like PDSI_China, this greater sensitivity to local climate improves its ability to reflect drought change in the short term compared with PDSI_China.

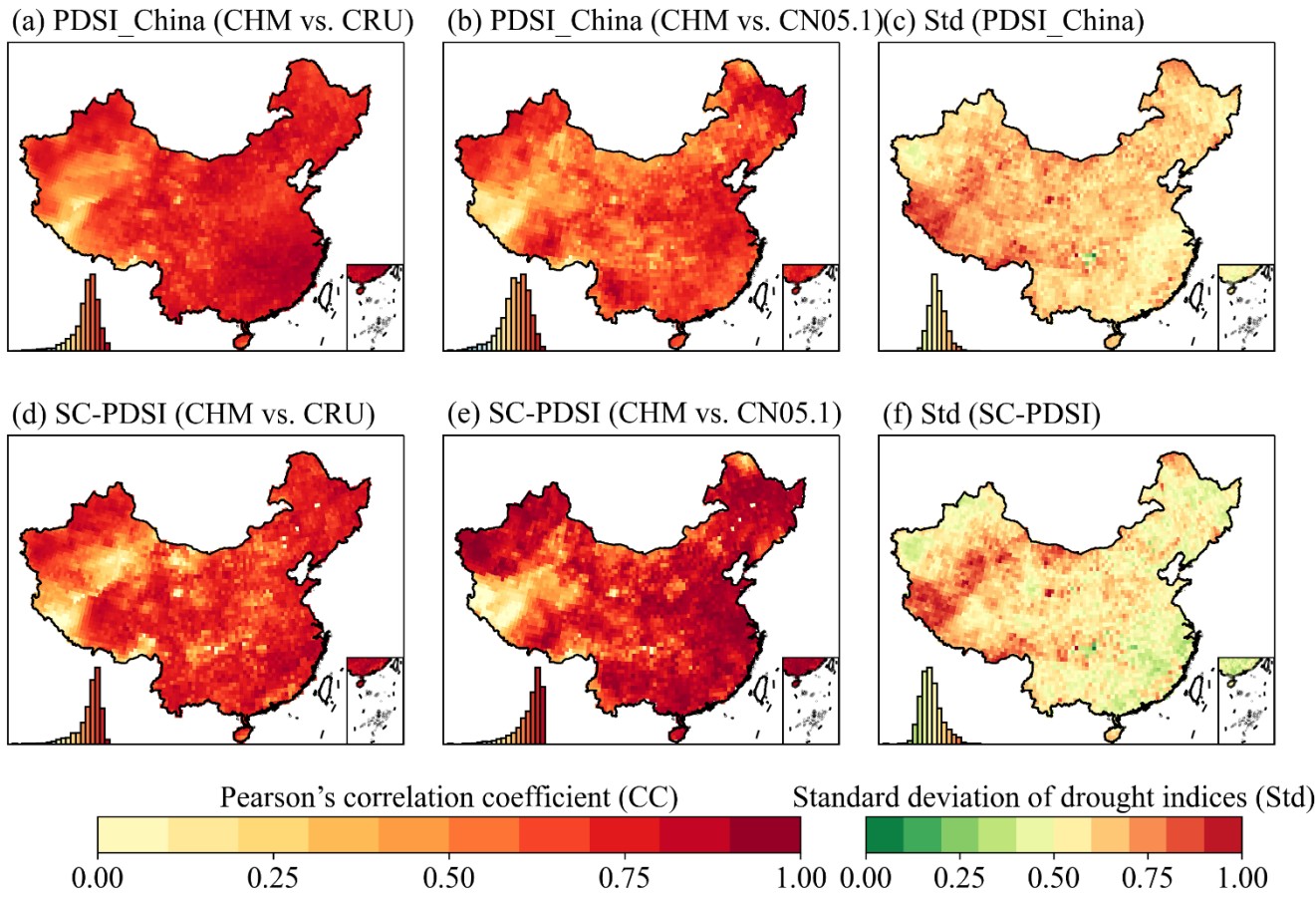

**Figure 8: (a–b) Correlation spatial distributions of PDSI_China and SC-PDSI based on CHM and CRU data. (d–e) Correlation spatial distributions of PDSI_China, and SC-PDSI based on CHM and CN05.1 data. (c) and (f) Spatial distribution of the standard deviations of PDSI_China and SC-PDSI across three data sources (CRU, CHM, and CN05.1).** The histogram at the bottom left in each subplot shows the distribution of values for all grid cells.

## 4.4 Consistency assessment of VPD based on CHM_Drought with CN05.1_Drought

When the VPD value is higher, it indicates that the atmosphere is drier, the transpiration of plants is enhanced, and more water is needed to maintain growth. Therefore, VPD reflects the water use demand of plants to a large extent. Figure 9 shows results from the consistency evaluation of VPD calculated using CHM and CN05.1. We found that the consistency of VPD calculated using CN05.1 is very high, and the correlation in each region of China is generally above 0.8 (Figure 9a). In addition, we compared the seasonal distribution of VPD with the results of Yuan et al. (2019) and found that the seasonal spatial distribution is also very consistent. It is mainly reflected in the high VPD in the arid and semi-arid areas of northwest China and the low VPD in the Tibetan Plateau, northeast China, and most regions of south China, especially in summer (Figure S6).

Studies have shown that VPD's impact on land productivity change in China is second only to soil moisture (Cheng et al., 2022), and the effect of high VPD is greater than that of high soil moisture in promoting vegetation productivity (Tu et al., 2024). Therefore, we believe that the correlation between NDVI and VPD (Figure S7) serves as an additional metric to evaluate the data. We found that the correlation between VPD and NDVI was lower in the arid northwest and southwest of China. This may be due to water limitation in the arid northwest. Due to the very limited precipitation in this region, vegetation growth and development may mainly depend on water availability rather than VPD. VPD mainly describes the dryness of the air. In extremely arid conditions, even if the VPD is high, vegetation growth may be severely limited by water scarcity, resulting in a lower correlation between VPD and NDVI. In contrast, southwest China's topography is complex, including high mountains and deep valleys, and the climate types are diverse. These topographic and climatic conditions may lead to a more complex relationship between VPD and NDVI. For example, mountainous areas may have lower VPD due to the frequent occurrence of clouds and fog, but this may not necessarily reflect the actual water status of the vegetation on the ground. Rainfall and clouds are common in southwest China and may reduce the dryness of the air, thus affecting the correlation between VPD and NDVI. In addition, different types of vegetation have different responses to VPD. For example, some plants can survive in arid conditions by regulating the opening of stomata to reduce water evaporation. In southwest China, diverse vegetation types (e.g., evergreen forest, shrub, and crops) may have different physiological responses to changes in VPD, which may lead to a lower correlation between VPD and NDVI.

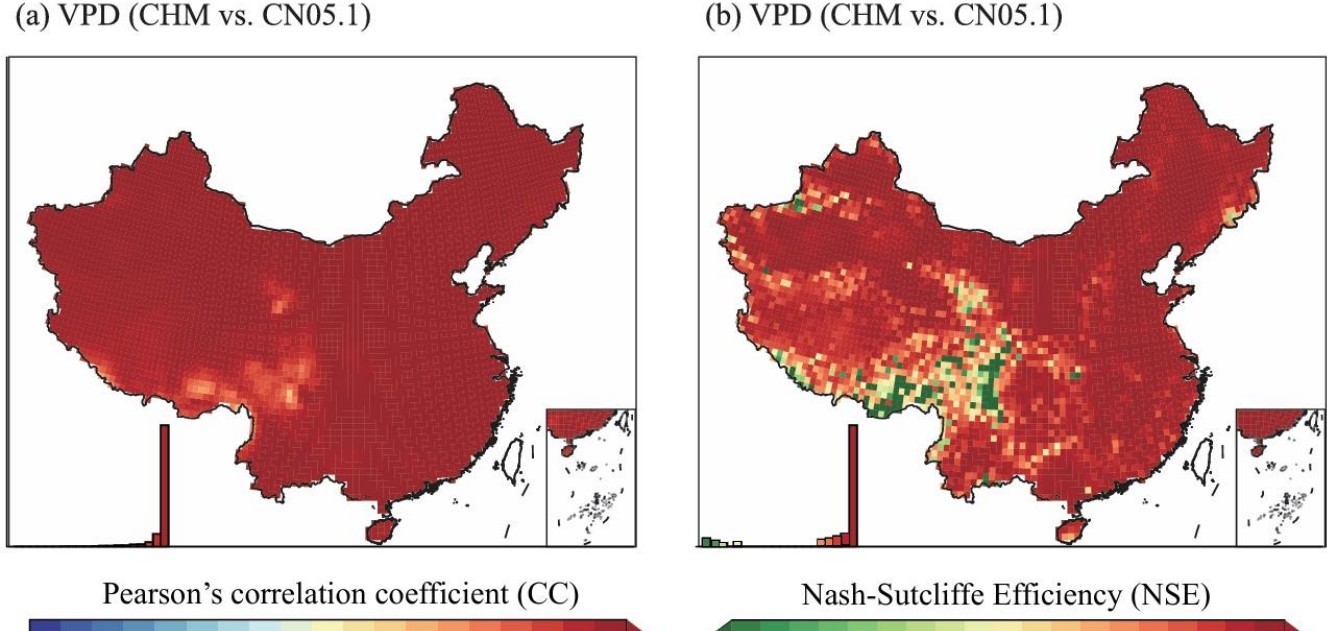

Figure 9: Spatial distributions of correlation (a) and NSE (b) of VPD based on CHM and CN05.1 data

## 5 Limitations and future work

Although this study provides valuable data resources for drought research in China, there are also limitations that point to potential directions for future research. First, the data uncertainty in northwest China is relatively large due to meteorological stations being sparse. Although both CRU and CN05.1 are obtained based on station interpolation, the data consistency is generally not high in areas with sparse meteorological stations (including hyper-arid areas from Figure 1) when compared with their calculated drought index. This may hinder the accurate assessment and understanding of drought conditions in the region.

Second, because the spatial resolution of CHM_Drought is 0.1°, the higher the accuracy, the lower the confidence of the spatial resolution in the site-sparse areas. However, it is important to recognize the inherent limitations of uncertainty quantification in sparse observational data. To address these uncertainties, future work could integrate satellite data or reanalysis products to expand spatial coverage, or apply advanced machine learning techniques, such as deep learning, to capture complex spatial and temporal patterns. Third, although six common meteorological drought indices have been developed and are powerful

tools for understanding the diversity of drought, PDSI_China and SC-PDSI, for example, are more dependent on the accuracy of soil available water capacity (AWC) data, and there are currently no high-quality AWC data with a spatial resolution matching that of CHM_Drought (0.1°).

Additionally, a critical area for future work involves the use of the latest climate projections, such as those from the Coupled Model Intercomparison Project Phase 6 (CMIP6), to estimate future values of drought indices. This approach could offer more

robust and detailed insights into how climate change may impact drought frequency, intensity, and duration in China and globally. Integrating CMIP6 projections with drought indices can help in understanding future drought risks under various greenhouse gas emission scenarios, thereby enhancing drought preparedness and mitigation strategies. Moreover, this could also involve developing or refining drought indices (including an agricultural drought index or hydrological drought index) that are more sensitive to projected changes in precipitation, temperature, evaporation, and other climatic variables influenced

by climate change.

## 6 Data availability

This high-resolution long-term drought dataset covers the period of 1961–2022, and it will continue to be updated annually. It contains data for spatial resolutions $0.1° \times 0.1°$ covering the domain of 18–54° N, 72–136° E. The NetCDF formatted output files of the CHM_Drought dataset are freely accessible at https://doi.org/10.6084/m9.figshare.25656951.v2 (Zhang and Miao,

465 2024).

## 7 Conclusions

We developed new high-resolution multi-drought indices from data across mainland China with a 0.1° resolution, spanning 1961 to 2022. The dataset includes six meteorological drought indices, namely SPI, SPEI, EDDI, PDSI_China, and SC-PDSI. All six drought indices can monitor drought events in China well, although different indices and different scales have different

performance characteristics. The shorter timescale (2-week) drought index can be used as an early warning tool for drought, but it is more sensitive to short-term precipitation or temperature, which may limit its use in monitoring drought or cold areas. The developed dataset CHM_Drought is highly consistent with the drought indices calculated on the basis of CRU and CN05.1. In conclusion, the development of this high-resolution (0.1°), reliable drought dataset for China from 1961 to 2022 marks a multifaceted contribution to drought research and management. It not only enhances our ability to monitor, predict, and respond

to drought conditions but will also support strategic planning across multiple sectors, including agricultural planning and management, water resources management, climate change adaptation strategies, and even interdisciplinary research to enable researchers to understand the compounding effects of drought. By addressing the urgent need for accurate and accessible drought data, this dataset opens new avenues for research and policy-making that can mitigate the impacts of drought and contribute to the sustainable management of natural resources.

**Author contributions**

QZ and CM contributed to designing the research; QZ implemented the research and wrote the original draft; CM supervised the research; all co-authors revised the manuscript and contributed to the writing.

**Competing interests**

The contact author has declared that none of the authors has any competing interests.

**Acknowledgments**

This research was supported by the National Key Research and Development Program of China (No. 2024YFF0809301) and the National Natural Science Foundation of China (No. U24A20572). We are also grateful to the National Meteorological Information Center of the China Meteorological Administration (NMIC, http://data.cma.cn) for providing the observed climate data.

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
