# Peer review of "A New High-Resolution Multi-Drought Indices Dataset for Mainland China"

_Earth System Science Data, 2024_

## Author Response (AR1)

**A DETAILED LIST OF RESPONSES**
**TO THE EDITOR**

We greatly appreciate your careful reading of the manuscript, insightful comments, and valuable suggestions. Your thoughtful review has enhanced our paper considerably. The manuscript has been revised thoroughly according to your comments and those of the individual reviewers, with our point-by-point responses detailed below.

(1) When preparing the response, the authors should be careful to address the reviewers comments. I recommend to take particular care about the question regarding uncertainty quantification of the drought indicators. This is of particular importance given the fact that there is a low station density in western half of the dataset (especially Qinghai-Tibetan Plateau and Xinjiang).

**Response:** Many thanks for your comments. Two reviewers both mentioned the problem of quantification of uncertainty: one suggests making each index generate an uncertainty graph, and the other suggests we discuss whether uncertainty is caused by the sparseness of sites or by interpolation methods. We have responded to these two reviewers in detail. Let's summarize here: (1) We used standard deviation alone to quantify the uncertainty of the drought index in different data sources (Figure S4). In Figure S4, we found that the region with greatest uncertainty was mainly the northwestern region, with its low density of meteorological stations. (2) The accuracy of different interpolation methods was more consistent in regions with uniform distribution and high density of meteorological stations. This has been demonstrated in previous work by our team (Han et al., 2023) In areas with sparse stations, the ADW method is slightly better than other methods due to its consideration of distance and angular weighting (Han et al., 2023). Refer to Question 6 from Reviewer #1 and Question 4 from Reviewer #2 for our specific responses, and for the selection of the ADW method, refer to Question 2 from Reviewer #1.

(2) I also have a question about the comparison to CRU on a 0.5x0.5 degree grid. It seems that the figures displaying this have some interpolation/ smoothing indicating a

higher resolution than stated in the manuscript, which should be either removed or clearly explained.

**Response:** Many thanks for your comments. For visualization, we had used the python method *plot.contourf*, which does have a smoothing and beautifying effect when drawing, and we have now redrawn all the figures to show the original spatial resolution.

References:

Han, J., Miao, C., Gou, J., Zheng, H., Zhang, Q., and Guo, X.: A new daily gridded precipitation dataset for the Chinese mainland based on gauge observations, Earth Syst Sci Data, 15, https://doi.org/10.5194/essd-15-3147-2023, 2023.

---------------------------------------------- end line -----------------------------------------

For your convenience, to make the review of our revisions easier, we have marked all responses and related revisions in light blue.

**A DETAILED LIST OF RESPONSES**
**TO REVIEWER #1**

We greatly appreciate your careful reading of the manuscript, insightful comments, and valuable suggestions. Your thoughtful review has enhanced our paper considerably. The manuscript has been revised thoroughly according to your comments, with our point-by-point responses detailed below.

(1) Section 2.2: It is suggested to elaborate missing data handling, as it is one of the tricky part of the observational data while considering several meteorological parameters.

**Response:** Many thanks for your comments. The meteorological station data we used was from the China Meteorological Administration (CMA; http://data.cma.cn/), from 1961 to 2022. These data have missing values at different time periods, and we did not fill in the time series for the missing values, because any filling method will introduce errors.

To avoid potential human errors in filling in missing data in the time series, we exclusively perform spatial interpolation on the observational data, using only the available data.

Here, we provide an example using temperature data. Only stations available each day are used for interpolation, despite daily variation. In Fig. R1, red dots indicate stations with increased data compared to the previous decade, showing a growing number of stations available for spatial interpolation over time. In Section 2.2 (Lines 152-153) we added the following: "For missing values, we did not fill in the time series, but used only stations with available data for spatial interpolation each day."

[Figure]

Fig. R1. Spatial distribution of average temperature (Tmean) monitoring stations over time. Black dots represent stations that existed in the previous time period, while red dots indicate stations newly added compared to the earlier time frame.

(2) Why angular distance-weighted interpolation (ADW) is considered for the higher-resolution gridding? Is it better than optimum interpolation method and state-of-the-art objective analysis techniques? Considering variability of meteorological parameters taken in this study, does ADW reasonable for all parameters?

**Response:** Many thanks for your comments. There are indeed many spatial interpolation methods. Our team's previous work has proved that the ADW method is the most suitable for precipitation interpolation (Han et al., 2023). The advantage of ADW is that it takes into account both the distance and angular relationships between stations, making it more robust in areas where data points are sparsely or irregularly distributed. This dual weighting system allows for more accurate interpolation in complex geographical settings, providing more reliable results for high-resolution gridding. Second, to maintain the consistency of the interpolation method, we apply the ADW method to all variables and consider the correlation decay distance (CDD) for each variable. ADW with CDD provides a key benefit that other methods may not

emphasize as directly: the gradual reduction of correlation with increasing distance between stations. By explicitly modeling the decay of correlation between station pairs (e.g., as illustrated for all variables in Fig. R2-R7), CDD ensures that distant stations contribute less to the interpolation, while nearby, highly correlated stations are given more weight. This distance-weighting characteristic is critical in regions with uneven station distributions, where ADW can mitigate the influence of distant stations that may not reflect local conditions accurately. In Section 2.2 (Lines 148-152) we added the following: "Before calculating the drought index, we interpolated the basic meteorological variables (Tmax, Tmin, Tmean, Wind, Ssd, Rh; see Figure 2) and considered the correlation decay distance (CDD) for each variable, and in the interpolation process we adopted angular distance–weighted interpolation (ADW), which considers angular weight in addition to the distance weight function, making it more robust to outliers. ADW with CDD provides a key benefit that other methods may not emphasize as directly: the gradual decrease of correlation with increasing distance between stations."

Compared to the optimal interpolation (OI) method, ADW offers a simpler computational process suited for fast processing of large-scale data, making it ideal for real-time applications. ADW is particularly effective in regions with complex terrain, such as China, where it can handle spatial heterogeneity and provide smoother transitions across varying geographic and climatic conditions. Studies show that ADW more accurately captures spatial variations and reduces interpolation error in meteorological data. For instance, Chen et al. (2024) generated a high-accuracy raster dataset of China's extreme temperature indices (1961–2020) using ADW, verifying its strong applicability in China.

[Figure]

Fig. R2. Estimation of correlation decay distance (CDD ≈ 99) for daily mean temperature (Tmean) series for all stations in the interpolated domain. Black points show the distance–correlation pair for each station. The red line is the exponential curve fitted to the data by ordinary least squares. The blue dashed line marks where correlation equals $1/e$.

[Figure]

Fig. R3. Estimation of correlation decay distance (CDD ≈ 136) for daily min temperature (Tmin) series for all stations in the interpolated domain. Black points show the distance–correlation pair for each station. The red line is the exponential curve fitted to the data by ordinary least squares. The blue dashed line marks where correlation equals $1/e$.

[Figure]

Fig. R4. Estimation of correlation decay distance (CDD $\approx$ 272) for daily max temperature (Tmax) series for all stations in the interpolated domain. Black points show the distance–correlation pair for each station. The red line is the exponential curve fitted to the data by ordinary least squares. The blue dashed line marks where correlation equals $1/e$.

[Figure]

Fig. R5. Estimation of correlation decay distance (CDD≈361) for daily wind speed (Wind) series for all stations in the interpolated domain. Black points show the distance–correlation pair for each station. The red line is the exponential curve fitted to the data by ordinary least squares. The blue dashed line marks where correlation equals $1/e$.

[Figure]

Fig. R6. Estimation of correlation decay distance (CDD≈480) for daily sunshine duration (Ssd) series for all stations in the interpolated domain. Black points show the distance–correlation pair for each station. The red line is the exponential curve fitted to the data by ordinary least squares. The blue dashed line marks where correlation equals $1/e$.

[Figure]

Fig. R7. Estimation of correlation decay distance (CDD ≈ 420) for daily average relative humidity (Rh) series for all stations in the interpolated domain. Black points show the distance–correlation pair for each station. The red line is the exponential curve fitted to the data by ordinary least squares. The blue dashed line marks where correlation equals $1/e$.

(3) Why authors have not considered multivariate drought indices including precipitation and soil moisture? An example of such long-term global datasets can be found at https://doi.org/10.1088/1748-9326/7/4/044037

**Response:** Many thanks for your comments. Considering that different drought indices can characterize different types of drought, and that there are so many drought indices to date, in this work we focus only on meteorological drought, which is a precursor to other types of drought. Accurate analysis of meteorological droughts is essential for predicting and understanding the development of other droughts, such as agricultural and hydrological droughts. By first conducting in-depth research on meteorological drought, we can lay the foundation for subsequent research on multiple drought types,

and improve the accuracy of overall drought monitoring and early warnings. Considering a multivariate drought index could indeed allow more comprehensive drought analysis; however, other data (such as soil moisture data) are difficult to obtain, and there are shortcomings in data quality and spatiotemporal coverage. Therefore, we plan to gradually introduce and integrate multiple-drought-type data in future studies to achieve more comprehensive drought assessment.

Additionally, as noted in the work of Aghakouchak and Nakhjiri (2012), the integration of GPCP and satellite data facilitates robust monitoring of past and near-real-time meteorological drought conditions through SPI calculation; however, their approach focuses solely on precipitation. Our study, on the other hand, calculates multiple indices that require a broader range of meteorological variables. For example, the calculation of potential evapotranspiration (PET), as used in the SPEI, requires variables such as precipitation, temperature, sunshine duration, and wind speed. Moreover, most satellite products lack key variables such as relative humidity and sunshine duration, which is why we rely on high-quality interpolated meteorological data for our drought index calculations.

(4) Yangtze River basin may be highlighted in any one figure for the convenience of the global readers.

**Response:** Many thanks for your comments. CHM_Drought covers the entire country from 1961 to 2022. For demonstration, in Section 4.1 of the paper, we take the summer of 2022 in the Yangtze River Basin as an example to examine the monitoring capabilities of each drought index. In the original, we used national data and did not show the scope of the Yangtze River basin, so we revised the manuscript to include the Yangtze River basin boundary to aid international readers. Among them, SPI, SPEI, and EDDI have multi-scale characteristics, so we show SPI, SPEI, and EDDI at a three-month scale (June, July, August; JJA) in Figure 3, while PDSI_China, SC-PDSI, and VPD represent mean summer (JJA) values. In addition, since the drought in the Yangtze River Basin this summer was prolonged, persisting into autumn, different time scales of the drought index could be used to monitor the initial hot spot of the drought and the

drought conditions caused after a long accumulation time. Similarly, the boundary of the Yangtze River Basin was also included in Figure 4. In Section 4.1 (Lines 285-287) we added the following: "A severe drought occurred in the south of China in the summer of 2022, mainly concentrated in the Yangtze River basin. To show the performance of the CHM_Drought dataset in monitoring drought, we use the summer (June, July, August; JJA) of 2022 in the Yangtze River Basin as an example to examine the monitoring capabilities of drought indices."

[Figure]

Figure 3: Spatial distribution of summer (June, July, August; JJA) drought characteristics in the Yangtze River Basin, China. Here, (a), (b), and (c) depict the three-month scale spatial distribution of drought indices, while (d), (e), and (f) present the average summer (JJA) values for these indices.

[Figure]

Figure 4: Spatial distribution of three drought indices (SPI, SPEI, and EDDI) in the Yangtze River Basin, China, across multiple timescales (2-week, 1-month, 3-month, and 6-month), using August 2022 as an example. (a–d) SPI-2W shows the 2-week scale SPI, SPI-1 shows the 1-month scale SPI, SPI-3 shows the 3-month scale SPI, and SPI-6 shows the 6-month scale SPI. The scales of SPEI and EDDI follow the same naming pattern.

(5) How empirical constant of expression 8 was determined. It needs to be elaborated.

**Response:** Many thanks for your comments. We used the modified calculation method for China-specific PDSI provided by the China National Standard for Meteorological Drought Classification (Standard No. GB/T 20481-2017; hereafter referred to as GB/T) (Zhong et al., 2019) for the calculation of PDSI_China, which includes the empirical constant in Expression 8.

Comparisons done for 9 stations in 7 states by Palmer. (1965) indicated different $\sum \bar{D}_i K_i'$ values, and the average value of $\sum \bar{D}_i K_i'$ for these 9 stations (i.e., 17.67) was

taken as the numerator and the $\sum \overline{D}_J K_i'$ for a given region was taken as the denominator. GB/T sets *a* in expression 8 to 16.84 instead of 17.67 according to the results of An et al. (1985). Specifically, An et al. (1985) selected the relevant data from 12 meteorological stations (Beijing, Qingdao, Xian, Xuzhou, Hohhot, Taiyuan, Hanzhong, Jiamusi, Shenyang, Hankou, Wuzhou, and Kunming) to revise the Palmer drought degree model in the process of revising the weight factors. For the detailed formula, please refer to the supplementary document.

$$K_i = \frac{a}{\sum_{j=1}^{12} \overline{D}_J K_j'} K_i' \tag{8}$$

(6) What is the role of land use/ land cover on drought indices? Is it possible to introduce any new index considering land use/land cover change?

**Response:** Many thanks for your comments. Although land use and land cover (LULC) significantly impacts drought, particularly in agricultural and hydrological contexts, our focus is on meteorological variables such as precipitation and temperature. On the other hand, because our high-resolution drought index is based on observational data, meteorological data already reflect land-atmosphere interactions, including LULC changes. Thus, the effect of LULC is inherent, although not explicitly isolated in this study.

Our current priority is to develop the CHM_Drought dataset, focused on high-resolution meteorological drought indices across mainland China from 1961 to 2022. Integrating LULC changes for a more comprehensive drought index is a valuable goal for future research.

(6) It is suggested to prepare an uncertainty map for each drought index. It would be vital for end users.

**Response:** Many thanks for your comments. We quantified the uncertainty of each indexusing standard deviation (Figure S4). Among the 6 drought indices involved in the study, the CRU data lacked relative humidity variables when calculating VPD index, which could not be calculated. Therefore, standard deviation was calculated for

all indices except the VPD index (Figure S4). In Section 4.2 (Lines 363-365) we added the following: "In addition, we quantified the uncertainties of SPI, SPEI and EDDI at different time scales (Figure S4). We used standard deviation to quantify the results, which were similar to those in Figures S1–S3. The results all showed that the highest uncertainties were mainly found in areas with few stations."

[Figure]

Figure S4: Spatial distribution of the standard deviations of the SPI, SPEI, and EDDI drought indices across three data sources (CRU, CHM, and CN05.1) at various time scales (1-, 3-, 6-, and 12-month). Here, (a-c) show the 1-month scale, (d-f) show the 3-month scale, (g-i) show the 6-month scale, and (j-l) show the 12-month scale.

References:

An, S., and Xing, J.: Modified Palmer drought index and its application, Atmosphere,1985, (12):17-19 (in Chinese).

Aghakouchak, A. and Nakhjiri, N.: A near real-time satellite-based global drought climate data record, Environmental Research Letters, 7, https://doi.org/10.1088/1748-9326/7/4/044037, 2012.

Chen, Q., Zhang, Y., Liu, X., Lian, Q., and Xu, J.: Development of gridded dataset of extreme temperature index in China based on ETCCDI. Journal of Global Change Data & Discovery, 2024, 8(1): 67–75. https://doi.org/10.3974/geodp.2024.01.08.

Han, J., Miao, C., Gou, J., Zheng, H., Zhang, Q., and Guo, X.: A new daily gridded precipitation dataset for the Chinese mainland based on gauge observations, Earth Syst Sci Data, 15, https://doi.org/10.5194/essd-15-3147-2023, 2023.

Palmer, W. C., 1965: Meteorological drought. Office of Climatology Research Paper 45, Weather Bureau, Washington, D.C., 58 pp.

Zhong, R., Chen, X., Lai, C., Wang, Z., Lian, Y., Yu, H., and Wu, X.: Drought monitoring utility of satellite-based precipitation products across mainland China, J Hydrol, 568, https://doi.org/10.1016/j.jhydrol.2018.10.072, 2019.

----------------------------------------------- end line -----------------------------------------

For your convenience, to make the review of our revisions easier, we have marked all responses and related revisions in light blue.

**A DETAILED LIST OF RESPONSES**

**TO REVIEWER #2**

We greatly appreciate your careful reading of the manuscript, insightful comments, and valuable suggestions. Your thoughtful review has enhanced our paper considerably. The manuscript has been revised thoroughly according to your comments, with our point-by-point responses detailed below.

(1) Many datasets are used in this study, while their description in this section is unclear. It would be better to provide a clear classification of these data sources and distinguish between the meteorological data from CMA, CHM, and CN05.1, as all the three datasets seem to originate from gauge observations in China. Further, the CRU dataset is based on global gauge observations with fewer gauges in China. The authors would clarify why these datasets are included in the consistency test.

**Response:** Many thanks for your comments. In this study, we used multiple datasets. To clarify their usage, in Section 2.1 (Lines 117-127) we added the following:

"We used several datasets, including the daily meteorological station data (Figure 1) from the China Meteorological Administration (CMA; http://data.cma.cn/), gridded precipitation data from CHM_PRE (Han et al., 2023; https://data.tpdc.ac.cn/zh-hans/data/e5c335d9-cbb9-48a6-ba35-d67dd614bb8c), and data from both CRU (https://crudata.uea.ac.uk/cru/data/hrg/) and CN05.1 (a gridded daily observation dataset over mainland China; https://ccrc.iap.ac.cn/resource/detail?id=228). First, we applied meteorological station data from the CMA to interpolate basic meteorological variables from 1961 to 2022 with spatial resolution of 0.1°, including maximum temperature (Tmax), minimum temperature (Tmin), mean temperature (Tmean), average wind speed (Wind), sunshine duration (Ssd), and average relative humidity (Rh). We directly used CHM_PRE and the interpolated meteorological data to compute CHM_Drought. The CN05.1 and CRU datasets were collected to evaluate CHM_Drought, with CRU data covering Precapitation (Pre), Tmax, Tmin, Tmean, Wind, and Ssd, and CN05.1 data covering Pre, Tmax, Tmin, Tmean, Wind, Rh, and

Ssd. Notably, CN05.1's Ssd data spans 1961 to 2018, while other variables span 1961 to 2022."

Additionally, CRU was chosen for the consistency test because it is among the most widely used global datasets. Although CRU's station coverage within China is limited, it remains a standard in international climate research, and it is widely applied in global-scale meteorological and drought analysis. Including CRU allows us to validate the consistency of CHM_Drought under global standards. Moreover, using CRU enables a diversity assessment across different data sources; as a comprehensive source of global meteorological station data, CRU provides a different observational structure than domestic Chinese stations. By incorporating CRU data for comparison, we can examine CHM_Drought's performance across varied data sources and observational distributions, further evaluating its consistency and adaptability in conditions of sparse or uneven data distribution.

(2) FAO-56 Penman-Monteith equation is designed to define the reference crop ET (ET0) using a hypothetical reference crop with an assumed height of 0.12 m. Here, the authors used this equation to calculate PET rather than ET0. I suggest they provide an explanation of PET and ET0 and clarify the calculations used for each.

**Response:** Many thanks for your comments. In our study, EDDI and SPEI were calculated based on $ET_0$ and PET, respectively, and this choice was mainly based on the choice of the developers of the EDDI index and the SPEI index. Specifically, Hobbins et al. (2016) developed EDDI to focus on atmospheric evaporation requirements to characterize drying potential. Reference crop evapotranspiration (ET0), based on hypothetical reference crop conditions (height of 0.12 m), is designed to reflect the physical atmospheric demand independent of surface water availability. The physical basis of ET0 eliminates the need for remote sensing data to estimate atmospheric evaporative demand, allowing it to consistently capture the intensity of atmospheric 'thirst' for moisture and effectively indicate vegetation water stress. This

makes EDDI a reliable metric for monitoring atmospheric drying potential, particularly under rapid or sustained drought conditions.

Unlike EDDI, the standardized precipitation evapotranspiration index (SPEI) developed by Vicente-Serrano et al. (2010) considers not only atmospheric demand but also surface water availability. Potential evapotranspiration (PET) is generally applied under conditions of adequate water supply, representing the maximum potential for evapotranspiration. SPEI uses PET to assess the balance between precipitation and evaporative demand, enabling it to capture water imbalances associated with drought. PET reflects the influence of broader climatic variables on water supply and demand, making it well-suited for SPEI's comprehensive evaluation of water balance.

In summary, EDDI uses $ET_0$ to focus on changes in atmospheric evaporative demand, independent of water supply constraints, allowing for a rapid indication of drying potential. In contrast, SPEI uses PET to capture the dynamic balance between precipitation and evaporative demand, providing a more comprehensive depiction of water imbalances associated with drought.

(3) why was August 2022 chosen as the node of the 2022 severe drought in the Yangtze River basin? In fact, this drought lasted from summer to autumn. The cumulative water shortage in the months following August may be worse.

**Response:** Many thanks for your comments. The August we described in Section 4.1 is not precise; we intended to describe the summer of 2022. The three-month drought index values for August actually represent the cumulative drought conditions for June, July, and August (JJA, summer). Accordingly, we have revised the chart and updated the chart notes (see Figure 3 and 4). For drought indices without multi-scale features (PDSI_China, SC-PDSI and VPD), we show the mean values in the summer of 2022 in Figure 3. In Section 4.1 (Lines 285-287) we added the following: "A severe drought occurred in the south of China in the summer of 2022, mainly concentrated in the Yangtze River basin. To show the performance of the CHM_Drought dataset in monitoring drought, we use the summer (June, July, August; JJA) of 2022 in the Yangtze River Basin as an example to examine the monitoring capabilities of drought

indices. For SPI, SPEI, and EDDI we selected a 3-month scale (seasonal scale; Jin et al., 2020), as shown in Figure 3."

On the other hand, although this drought extended from summer into autumn, with cumulative water shortages potentially becoming more severe at some grid points after August, summer represents a crucial turning point in the 2022 Yangtze River Basin drought event. It marks the peak in water deficit and atmospheric evaporative demand. Additionally, due to data processing and presentation constraints, we selected summer as the sample node.

[Figure]

Figure 3: Spatial distribution of summer (June, July, August; JJA) drought characteristics in the Yangtze River Basin, China. Here, (a), (b), and (c) depict the three-month scale spatial distribution of drought indices, while (d), (e), and (f) present the average summer (JJA) values for these indices.

[Figure]

Figure 4: Spatial distribution of three drought indices (SPI, SPEI, and EDDI) in the Yangtze River Basin, China, across multiple timescales (2-week, 1-month, 3-month, and 6-month), using August 2022 as an example. (a–d) SPI-2W indicates the 2-week scale SPI, SPI-1 indicates the 1-month scale SPI, SPI-3 indicates the 3-month scale SPI, and SPI-6 indicates the 6-month scale SPI. The scales of SPEI and EDDI follow the same naming pattern.

(4) the low consistency between CHM_Drought with CN05.1_Drought is attributed to the poor performance of sparse sites. Does this imply that the data processing method (e.g., interpolation method) affects the accuracy of the production of drought datasets?

**Response:** Many thanks for your comments. Our team's previous research (Han et al., 2023) compared different interpolation methods in precipitation interpolation and found that ADW was the most suitable for precipitation interpolation. Meanwhile, it was found that different interpolation methods had higher accuracy in areas with uniform distribution and dense meteorological stations, and the differences between

them were small. On the contrary, in areas with sparse sites, all interpolation methods have large biases, including ADW. In addition, we use standard deviation to quantify the consistency of the drought index (see Figure S4), and we find that the regions with poor consistency are mainly those with sparse sites, so we find that sparseness of sites is the main cause of poor consistency.

[Figure]

Standard deviation of drought indices between CRU, CHM, and CN05.1

**Figure S4**: Spatial distribution of the standard deviations of the SPI, SPEI, and EDDI drought indices across three data sources (CRU, CHM, and CN05.1) at various time scales (1-, 3-, 6-, and 12-month). Where (a-c) show the 1-month scale, (d-f) show the 3-month scale, (g-i) show the 6-month scale, and (j-l) show the 12-month scale.

(5) While the correlation between VPD and NDVI is discussed, NDVI is influenced by various factors beyond VPD, and NDVI data itself may contain uncertainties. It is unclear why the correlation between VPD and NDVI can be used for the consistency assessment of VPD

**Response:** Many thanks for your comments. According to your suggestion, we have moved the correlation results for VPD and NDVI to the supplementary file. Although NDVI is affected by multiple factors such as precipitation, soil moisture, and land use, we retain the correlation chart between NDVI and VPD because previous studies have shown that VPD's impact on land productivity change in China is second only to soil moisture (Cheng et al., 2022), and the effect of high VPD is greater than that of high soil moisture in promoting vegetation productivity (Tu et al., 2024). Therefore, we believe that the correlation between NDVI and VPD serves as an additional metric to evaluate the data. In Section 4.4 (Lines 398-401) we added the following: "Studies have shown that VPD's impact on land productivity change in China is second only to soil moisture (Cheng et al., 2022), and the effect of high VPD is greater than that of high soil moisture in promoting vegetation productivity (Tu et al., 2024). Therefore, we believe that the correlation between NDVI and VPD (Figure S7) serves as an additional metric to evaluate the data."

In addition, the correlation between VPD and NDVI can help verify whether VPD is reflecting the drought response of vegetation. This approach is not strictly a conformance assessment, but rather a complementary validation to examine the correlation between VPD and vegetation water status.

(6) Some in-text citations are not listed in the Reference section, the authors should check it out.

**Response:** Many thanks for your comments. We checked the references carefully and made revisions.

References:

Cheng, Y., Liu, L., Cheng, L., Fa, K., Liu, X., Huo, Z., and Huang, G.: A shift in the dominant role of atmospheric vapor pressure deficit and soil moisture on vegetation greening in China, J Hydrol, 615, https://doi.org/10.1016/j.jhydrol.2022.128680, 2022.

Han, J., Miao, C., Gou, J., Zheng, H., Zhang, Q., and Guo, X.: A new daily gridded precipitation dataset for the Chinese mainland based on gauge observations, Earth Syst Sci Data, 15, https://doi.org/10.5194/essd-15-3147-2023, 2023.

Hobbins, M. T., Wood, A., McEvoy, D. J., Huntington, J. L., Morton, C., Anderson, M., and Hain, C.: The evaporative demand drought index. Part I: Linking drought evolution to variations in evaporative demand, J Hydrometeorol, 17, 1745–1761, https://doi.org/10.1175/JHM-D-15-0121.1, 2016.

Tu, Y., Wang, X., Zhou, J., Wang, X., Jia, Z., Ma, J., Yao, W., Zhang, X., Sun, Z., Luo, P., Feng, X., and Fu, B.: Atmospheric water demand dominates terrestrial ecosystem productivity in China, Agric For Meteorol, 355, 110151, https://doi.org/10.1016/J.AGRFORMET.2024.110151, 2024.

Vicente-Serrano, S. M., Beguería, S., and López-Moreno, J. I.: A multiscalar drought index sensitive to global warming: The standardized precipitation evapotranspiration index, J Clim, 23, 1696–1718, https://doi.org/10.1175/2009JCLI2909.1, 2010.

---------------------------------------------- end line ----------------------------------------

For your convenience, to make the review of our revisions easier, we have marked all responses and related revisions in light blue.

---

## Author Response (AR2)

**A DETAILED LIST OF RESPONSES**
**TO THE EDITOR**

We greatly appreciate your careful reading of the manuscript, insightful comments, and valuable suggestions. Your thoughtful review has enhanced our paper considerably. The manuscript has been revised thoroughly according to your comments and those of the individual reviewers, with our point-by-point responses detailed below.

(1) Given the importance of uncertainty quantification for the drought dataset, I think that Figure S4 should be added to the main body of the manuscript AND that there should be a suitable discussion around its interpretation. Such a discussion would ALSO include general limitations of uncertainty quantification in the presence of sparse observations.

**Response:** Many thanks for your comments. We have put the uncertainty graph (form Figure S4 to Figure 7) in the manuscript and discussed it around three main aspects: interpolation method uncertainty, interpolation station density, and consideration of covariates. Specifically, in Section 4.2 (Lines 371-390) we added the following: "In addition, we quantified the uncertainties of SPI, SPEI and EDDI at different time scales (Figure 7). We used standard deviation to quantify the results, which were similar to those in Figures S2–S4. The regions with higher standard deviation, such as the arid northwest, highlight the spatial variability in uncertainty across different datasets. This suggests that the drought indices calculated from these datasets may show obvious discrepancies in regions with sparse observational coverage. These uncertainties may have the following reasons: (1) The variability in interpolation techniques across datasets is a critical factor contributing to uncertainty. For instance, the CHM dataset employs advanced interpolation techniques based on high-density observational stations, while the CRU and CN05.1 datasets utilize thin plate smooth spline (TPSS) and inverse distance weighting (IDW) methods, respectively (Harris et al., 2020; Xu et al., 2009). These methodological differences become particularly pronounced in areas with complex topography, such as the arid northwest. Xu et al. (2022) demonstrated that TPSS performs well in capturing broad climate gradients, it may overly smooth the

results in data-sparse regions, leading to underestimation of extremes. Conversely, IDW might overemphasize local station values, causing biases in interpolated fields (Shen et al., 2023). (2) Sparse observational coverage is another significant source of uncertainty. Liu et al. (2009) highlighted that the density of interpolation sites is the key factor influencing interpolation accuracy. They found that the performance of interpolation methods, such as kriging or IDW, deteriorates significantly as the number of sites decreases. (3) Differences in the inclusion of auxiliary covariates, such as topography, land cover, or climate zones, further contribute to dataset discrepancies. The CHM dataset incorporates high-resolution digital elevation models (DEM) as covariates, while the CRU dataset primarily relies on planar spatial gradients without explicitly considering terrain effects (Harris et al., 2020). This leads to substantial differences in regions with complex orography."

(2) To increase transparency of the methods: Figures R2-R7 should be added to the Supplement. To do so they should be modified to (1) include measures of the fit of the exponential decay (e.g. variance explained) and (2) ideally changed to contour/ density plots to better visualize the distribution.

**Response:** Many thanks for your comments. We have added the contents of the CDD section to the supplementary document, including the CDD concept and method description, as well as the CDD density map with these meteorological variables added in Figure S1. Specifically, in Supplementary document (Lines 26-39) we added the following: "The ADW interpolation method used for this study was the modified Shepard's algorithm, which introduces the concept of correlation decay distance (CDD), also called correlation length scale or decorrelation length (Shepard, 1984; Dunn et al., 2020). The CDD is defined as the distance at which the correlation between one station and all other stations decays below $1/e$, approximately corresponding to the significance level of 0.05 for the correlation within large samples (Jones et al., 1997; Harris et al., 2020). The number of stations for interpolating the target grid cell is well constrained by the CDD, thus improving the interpolation precision (New et al., 2000; Mitchell and Jones, 2005; Hofstra and New, 2009). For every station, correlations (r) and distances

(x) for each variable are shown in Figure S1, and the ordinary least-squares method was used to fit an exponential decay function: $r = e^{-x/CDD}$, take the meteorological variable Wind (Figure S1a), for example, the estimated CDD is 361 km (95 % confidence interval: 361 km) at the 0.05 significance level."

[Figure]

**Figure S1:** Kernel density visualization of the Correlation Decay Distance (CDD) and the distribution for meteorological variables (Wind≈361, Ssd≈480, Rh≈420, Tmax≈ 272, Tmean≈ 99, Tmin≈ 136) for all stations within the interpolated domain. Black points show the distance–correlation pair for each station. The blue line is the exponential curve fitted to the data by ordinary least squares. The red dashed line marks where correlation equals 1/e.

(3) Technical:L125: PrecIpitation, L351 > no comma after THAT
**Response:** Many thanks for your comments. We have modified.

References:

Han, J., Miao, C., Gou, J., Zheng, H., Zhang, Q., and Guo, X.: A new daily gridded precipitation dataset for the Chinese mainland based on gauge observations, Earth Syst Sci Data, 15, https://doi.org/10.5194/essd-15-3147-2023, 2023.

Dunn, R. J. H., Alexander, L. V., Donat, M. G., Zhang, X., Bador, M., Herold, N., Lippmann, T., Allan, R., Aguilar, E., Barry, A. A., Brunet, M., Caesar, J., Chagnaud, G., Cheng, V., Cinco, T., Durre, I., de Guzman, R., Htay, T. M., Wan Ibadullah, W. M., Bin Ibrahim, M. K. I., Khoshkam, M., Kruger, A., Kubota, H., Leng, T. W., Lim, G., Li-Sha, L., Marengo, J., Mbatha, S., McGree, S., Menne, M., de los Milagros Skansi, M., Ngwenya, S., Nkrumah, F., Oonariya, C., Pabon-Caicedo, J. D., Panthou, G., Pham, C., Rahimzadeh, F., Ramos, A., Salgado, E., Salinger, J., Sané, Y., Sopaheluwakan, A., Srivastava, A., Sun, Y., Timbal, B., Trachow, N., Trewin, B., van der Schrier, G., VazquezAguirre, J., Vasquez, R., Villarroel, C., Vincent, L., Vischel, T., Vose, R., and Bin Hj Yussof, M. N. A.: Development of an updated global land in situ-based data set of temperature and precipitation extremes: HadEX3, J. Geophys. Res.-Atmos., 125, e2019JD032263, https://doi.org/10.1029/2019JD032263, 2020.

Jones, P. D., Osborn, T. J., and Briffa, K. R.: Estimating sampling errors in large-scale temperature averages, J. Climate, 10, 2548–2568, https://doi.org/10.1175/15200442(1997)010<2548:ESEILS>2.0.CO;2, 1997.

Harris, I., Osborn, T. J., Jones, P., and Lister, D.: Version 4 of the CRU TS monthly high-resolution gridded multivariate climate dataset, Sci. Data, 7, 109, https://doi.org/10.1038/s41597-0200453-3, 2020.

Hofstra, N. and New, M.: Spatial variability in correlation decay distance and influence on angular-distance weighting interpolation of daily precipitation over Europe, Int. J. Climatol., 29, 18721880, https://doi.org/10.1002/joc.1819, 2009.

Mitchell, T. D. and Jones, P. D.: An improved method of constructing a database of monthly climate observations and associated high-resolution grids, Int. J. Climatol., 25, 693–712, https://doi.org/10.1002/joc.1181, 2005.

New, M., Hulme, M., and Jones, P.: Representing twentiethcentury space–time climate variability. part II: development of 1901–96 monthly grids of terrestrial

surface climate, J. Climate, 13, 2217–2238, https://doi.org/10.1175/15200442(2000)013<2217:RTCSTC>2.0.CO;2, 2000.

Shepard, D. S.: Computer Mapping: The SYMAP Interpolation Algorithm, in: Spatial Statistics and Models, edited by: Gaile, G. L., and Willmott, C. J., Springer Netherlands, Dordrecht, 133–145, https://doi.org/10.1007/978-94-017-3048-8_7, 1984.

----------------------------------------------- end line -------------------------------------------

For your convenience, to make the review of our revisions easier, we have marked all responses and related revisions in light blue.